# Delayed generation of functional virus-specific circulating T follicular helper cells correlates with severe COVID-19

Meng Yu [1], Afandi Charles [1], Alberto Cagigi[1], Wanda Christ [2], Björn Österberg[1], Sara Falck-Jones[1], Lida Azizmohammadi[1], Eric Åhlberg[1], Ryan Falck-Jones [3,4], Julia Svensson[1], Mu Nie[1], Anna Warnqvist[5], Fredrika Hellgren[1], Klara Lenart [1], Rodrigo Arcoverde Cerveira [1], Sebastian Ols [1], Gustaf Lindgren[1], Ang Lin[1], Holden Maecker [6], Max Bell [3,4], Niclas Johansson[7,8], Jan Albert [9,10], Christopher Sundling [7,8], Paulo Czarnewski[11], Jonas Klingström [2,12], Anna Färnert[7,8], Karin Loré [1] & Anna Smed-Sörensen [1] ✉

Effective humoral immune responses require well-orchestrated B and T follicular helper (Tfh) cell interactions. Whether these interactions are impaired and associated with COVID-19 disease severity is unclear. Here, longitudinal blood samples across COVID-19 disease severity are analysed. We find that during acute infection SARS-CoV-2-specific circulating Tfh (cTfh) cells expand with disease severity. SARS-CoV-2-specific cTfh cell frequencies correlate with plasmablast frequencies and SARS-CoV-2 antibody titers, avidity and neutralization. Furthermore, cTfh cells but not other memory CD4 T cells, from severe patients better induce plasmablast differentiation and antibody production compared to cTfh cells from mild patients. However, virus-specific cTfh cell development is delayed in patients that display or later develop severe disease compared to those with mild disease, which correlates with delayed induction of high-avidity neutralizing antibodies. Our study suggests that impaired generation of functional virus-specific cTfh cells delays high-quality antibody production at an early stage, potentially enabling progression to severe disease.

Severe acute respiratory syndrome coronavirus 2 (SARS-CoV-2) infection causes coronavirus disease 2019 (COVID-19) with a broad spectrum of clinical outcome ranging from asymptomatic to severe disease, including life-threatening respiratory failure, and even fatal outcome[1,2]. While immunopathology is clearly a driver of COVID-19, knowledge about how immunological differences may dictate disease severity is still incomplete[3]. Understanding the nature of longitudinal immune responses in COVID-19 patients could aid in understanding and even predicting disease severity, as well as identifying therapeutics and more effective vaccines.

SARS-CoV-2-specific T and B cells as well as antibodies are detectable in COVID-19 patients during acute infection across disease severity from asymptomatic to severe outcome[4], and the levels are directly proportional to disease severity level[5,6]. T-follicular helper (Tfh) cells are crucial in orchestrating humoral immunity by supporting B-cell activation and antibody generation[7–10]. In human secondary lymphoid tissues, Tfh cells upregulate expression of the chemokine receptor CXCR5, allowing for the localization of Tfh cells to the germinal centers (GCs). There, Tfh cells provide help to B cells via inducible costimulatory molecule (ICOS), CD40L and IL-21, to facilitate

class switch recombination, somatic hypermutation to form high-affinity antibodies, and finally the generation of long-lived antibody-secreting B cells[11]. In fatal COVID-19 patients, GCs were absent in the spleen and lymph nodes and this was associated with impaired Tfh cell differentiation[12,13], indicating that loss of Tfh cells might lead to fatal outcome in COVID-19 patients. But what function Tfh cells have in non-fatal COVID-19 patients is still unclear since obtaining longitudinal lymph node samples for research is typically not feasible during ongoing infection.

Instead, a CXCR5[+] subset of CD4[+] memory T cells, named circulating Tfh (cTfh), has been identified in human peripheral blood, which shares phenotypic and functional properties of bona fide Tfh cells[14,15]. Studies have shown that human cTfh cells originate from lymph nodes and traffic into blood via the thoracic duct[16,17]. In ICOS-deficient or CD40L-deficient patients, the formation of GCs is severely impaired and consequently the numbers of cTfh cells is significantly reduced, further supporting the hypothesis that cTfh cells are GC-derived[18]. Human cTfh cells can be divided into cTfh1 (CXCR3[+]CCR6[-]), cTfh2 (CXCR3[-]CCR6[-]) and cTfh17 (CXCR3[-]CCR6[+]). cTfh1, cTfh2, and cTfh17 cells share the signature transcription factors and cytokines of Th1 (IFNγ and T-bet), Th2 (IL-4, IL-5, IL-13, and GATA-3) and Th17 (IL-17, IL-22, and ROR γt) cells, respectively[19–22]. In vitro studies have shown that human cTfh1 cells sufficiently support only memory B-cell differentiation, while cTfh2 and cTfh17 cells can activate naive B cells[22,23]. Thus, cTfh cells could be considered a surrogate of lymphoid Tfh cells.

Studies focused on COVID-19 convalescent individuals show the presence of SARS-CoV-2-specific cTfh cells that correlate with SARS-CoV-2-neutralizing antibody titers[24–26]. However, data on cTfh cells during acute SARS-CoV-2 infection across disease severity is very limited. Thevarajan et al. reported that the frequency of ICOS[+]PD-1[+] cTfh cells increased during acute infection in COVID-19 patients compared to convalescent or healthy donors[27,28]. Increased expression of CXCR5 and ICOS on SARS-CoV-2-specific CD4 T cells in mild to severe COVID-19 patients with acute infection has been reported, but none of the studies tested cTfh functionality directly[4,29]. Longitudinal, functional data on cTfh cells in COVID-19 patients across disease severity from acute infection to convalescence is still missing, and whether cTfh cell frequency and function correlate with or even predict COVID-19 disease severity is still unclear.

Here, we show the characteristics of cTfh cells in COVID-19 patients across mild to severe disease, longitudinally from acute infection to 3 and 8 months convalescence and compare the functionality of cTfh cells from acute COVID-19 patients across disease severity. We show that during acute SARS-CoV-2 infection virus-specific circulating cTfh cells expand with increasing disease severity. However, SARS-CoV-2-specific cTfh cell development is delayed in patients that display or later develop severe COVID-19 compared to patients with mild disease, which correlates with delayed induction of high-avidity neutralizing antibodies, potentially enabling progression to severe disease.

## Results

### Longitudinal sampling during SARS-CoV-2 infection
In total, 49 adults with PCR-confirmed SARS-CoV-2 infection were enrolled in the study. Longitudinal blood samples were collected multiple times during acute, symptomatic disease as well as during convalescence (median 3 and 8 months from symptom onset) (Fig. 1a, b). Identical samples from 20 age- and gender-matched prepandemic healthy controls (HCs) were used as baseline comparisons (Fig. 1a and Table 1). Disease severity was assessed daily in admitted patients and classified according to the respiratory domain of the sequential organ failure assessment (SOFA) score, with additional levels for non-admitted patients as mild cases. Patients were grouped based on their peak disease severity, which may differ from disease severity at the time of sampling (Fig. 1b). In this study, patients with mild, moderate, and

severe peak COVID-19 disease severity are hereafter referred to as mild, moderate and severe patients. Medical records from patient groups across disease severity were analyzed (Table 2). The distribution of age varied significantly across peak disease severity groups of COVID-19 patients ($P < 0.05$), as did the male sex ($P < 0.01$) and the BMI ($P < 0.01$) (Table 2), similar to what has been observed in other COVID-19 patient cohorts[1,30].

### cTfh cell abundance and activation are altered in severe COVID-19
To understand the longitudinal dynamics of cTfh cell characteristics in COVID-19 patients across disease severity, from acute disease to convalescence, PBMCs from 41 patients and 20 healthy controls were analyzed by flow cytometry (Supplementary Table 1). cTfh cells were identified as CXCR5[+] memory CD4[+] T cells and cTfh cell activation was determined by the expression of CD38 and ICOS[16] (Fig. 1c). cTfh cells were further differentiated based on CCR6 and CXCR3 expression to identify cTfh1 (CXCR3[+] CCR6[-]), cTfh2 (CXCR3[-] CCR6[-]) and cTfh17 (CXCR3[-] CCR6[+]) cells (Fig. 1c). Total cTfh cell frequencies were significantly lower in moderate and severe patients compared to mild during acute disease but normalized to similar levels as observed in healthy controls (dotted line) during convalescence (Fig. 1d). During acute disease, frequencies of cTfh1 and cTfh17 but not cTfh2 cells were significantly lower in severe patients than in mild patients, while there was no significant difference in frequency of cTfh1, cTfh2, and cTfh17 cells in patients across disease severity at the 3 and 8 months convalescence timepoints (Fig. 1d). In contrast, frequencies of activated CD38[+] ICOS[+] cTfh cells were significantly higher in severe and moderate patients than in mild patients during acute disease and normalized during both 3 and 8 months convalescence (Fig. 1d). When looking at cTfh subsets, there was no statistically significant differences in the frequency of activated cTfh1, cTfh2 or cTfh17 cells during acute disease (Fig. 1d). These results show that as disease severity increased during acute SARS-CoV-2 infection, cTfh cells were lower in frequency but a higher proportion were activated, suggesting that cTfh cell frequency and activation associate with disease severity in COVID-19 patients during acute disease.

### Frequencies of activated cTfh cells correlate with SARS-CoV-2 plasma antibody level and plasmablast frequencies in patients during acute COVID-19
To assess whether cTfh cell patterns associate with the humoral immune response, longitudinal plasma antibody levels against SARS-CoV-2 spike and RBD by ELISA as well as the blood plasmablast (PB) frequency were analyzed by flow cytometry (Supplementary Fig. 1 and Supplementary Table 2). In line with previous reports from us and others[31–33], we found that plasma IgA and IgG against SARS-CoV-2 spike and RBD proteins increased with increasing disease severity, during both acute disease and 3 months convalescence (Fig. 2a). Furthermore, plasma IgA against SARS-CoV-2 RBD and plasma IgG against SARS-CoV-2 spike and RBD positively correlated with the frequency of activated cTfh cells during acute disease (Fig. 2b). Frequencies of activated cTfh1, cTfh2, and cTfh17 cells positively correlated with plasma spike and RBD IgG, and frequencies of activated cTfh1 and cTfh17 cells positively correlated with plasma RBD IgA (Fig. 2c). However, frequencies of bulk cTfh1, cTfh2, or cTfh17 cells were not correlated with titers of plasma antibodies against SARS-CoV-2 (Supplementary Fig. 2). Frequencies of blood plasmablast in severe and moderate COVID-19 patients were significantly higher than that in mild patients during acute disease and normalized to frequencies comparable to healthy controls during convalescence (Fig. 2d). During acute disease, frequencies of blood plasmablast in patients across disease severity positively correlated with the frequency of activated cTfh cells in the same samples (Fig. 2e), including cTfh1, cTfh2, and cTfh17 cells (Fig. 2f).

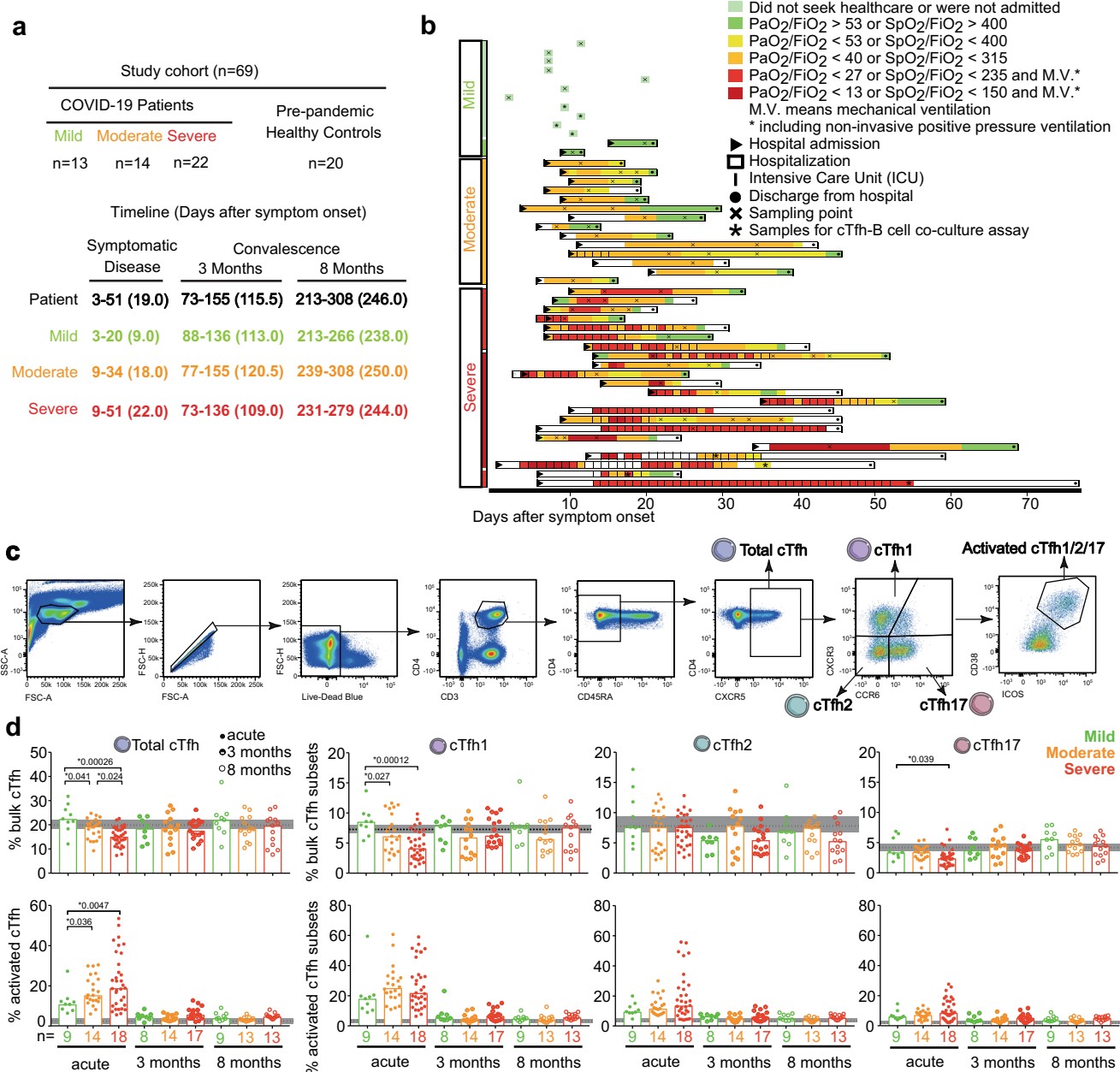

**Fig. 1 | Longitudinal frequency and activation of cTfh cells in COVID-19 patients across disease severity from acute disease up to 8 months convalescence.** **a** Overview of study cohort (n = 69), and **b** the timeline of longitudinal sampling. Patients are grouped based on peak disease severity, including mild (green), moderate (orange) and severe (red). Individual patients are color-coded based on daily disease severity. PaO2 is expressed in kPa. **c** Gating strategy to identify cTfh cells in PBMCs by flow cytometry. From single, live CD3⁺ CD4⁺ T cells, memory CD4⁺ T cells were identified as CD45RA⁻. From memory CD4⁺ T cells, cTfh cells were identified as CXCR5⁺ cells. cTfh subsets were identified as cTfh1 (CXCR3⁺ CCR6⁻), cTfh2 (CXCR3⁻ CCR6⁻) and cTfh17 (CXCR3⁻ CCR6⁺). Activated cTfh and its subsets were identified with ICOS⁺ CD38⁺ expressing. Patients are grouped based on peak disease severity, including mild (green), moderate (orange) and severe (red). **d** Bar charts show the frequency of bulk and activated cTfh, cTfh1, cTfh2, and cTfh17 cells from COVID-19 patients with acute disease (Acute, full circle), 3 months

convalescence (3 months, half circle) and 8 months convalescence (8 months, open circle) with median. Dots are individual samples color-coded according to peak disease severity. Dotted lines show the median frequency with 95% CI (gray area) of prepandemic healthy controls. *X* axis shows the number of patients in each bar. During acute disease, 9, 22, and 33 individual samples from 9 mild, 14 moderate and 18 severe patients, respectively, were analyzed using two-sided Generalized Estimating Equations (GEE) to account for the intra-person correlations inherent to repeated measures and assess statistically significant differences without adjusting for multiple comparisons. During 3 and 8 months convalescence, only one sample from each patient was analyzed and two-sided Kruskal–Wallis with Dunn's multiple comparisons test was used to assess statistically significant differences. *P* < 0.05 was considered to be a significant difference. *P values <0.05 are listed above each comparison. Source data are provided as a Source Data file.

## SARS-CoV-2 spike-specific and RBD-specific cTfh cell activation correlate with plasma antibody levels and plasmablast frequencies during acute COVID-19

Next, to mechanistically test whether cTfh cells are indeed reactive to SARS-CoV-2 antigens, we next measured the frequency of SARS-CoV-2 spike and RBD-specific cTfh cells in patients across disease severity

from acute infection to convalescence. PBMCs from 41 patients or 20 healthy controls were stimulated with SARS-CoV-2 spike or RBD proteins, or with BSA or SEB as negative and positive controls, respectively. After 20 h of stimulation, cTfh cells from COVID-19 patients but not cTfh cells from prepandemic healthy controls upregulated the activation markers CD25 and CD134 in response to the viral proteins (Fig. 3a and

**Table 1 | Demographic and clinical data of COVID-19 patients and healthy controls**

| | COVID-19 | Healthy controls | Significance[a] |
|---|---|---|---|
| Number of Individuals | 49 | 20 | |
| Age in year, median (range) | 55 (26–77) | 53 (24–81) | 0.4800 |
| Male, n (%) | 33 (67) | 14 (70) | >0.9999 |
| Female, n (%) | 16 (33) | 6 (30) | |
| **Comorbidities[b]** | | | |
| CCI, median (IQR) | 1 (3) | NA | NA |
| BMI, median (IQR) | 28.1 (5.4) | NA | NA |
| Hypertension, n (%) | 12 (24.5) | 0 | 0.0056 |
| Diabetes, n (%) | 13 (26.8) | 0 | 0.0056 |
| Current smoker, n (%) | 2 (4.1) | 0 | >0.9999 |
| **Laboratory analyses[b,c]** | | | |
| CRP[d], median (IQR) | 155 (174) | 1 (1) | <0.0001 |
| WBC[e], median (IQR) | 7.6 (4.9) | 6.8 (1.8) | 0.2730 |
| Neutrophils[e], median (IQR) | 6.2 (3.9) | 3.8 (1.3) | 0.0086 |
| Lymphocytes[e], median (IQR) | 0.8 (0.5) | 1.9 (0.7) | <0.0001 |
| NLR, median (IQR) | 7.4 (7.6) | 1.8 (1.3) | 0.0002 |

*CCI* Charlson Comorbidity Index, *NLR* neutrophil/lymphocyte ratio, *NA* data not available.

[a]Two-sided Mann–Whitney *U* and Fisher's exact tests were performed to determine statistical significance.

[b]Comorbidities and laboratory analyses data were available from 41 patients.

[c]Peak values: CRP. Nadir values: lymphocyte count. WBC and neutrophil count at the time point of the lowest lymphocyte count.

[d]mg/L.

[e]$10^9$ cells/L.

Normal range: CRP < 3 mg/L, WBC 3.5 ×$10^9$/L to 8.8 × $10^9$/L, lymphocytes 1.1 × $10^9$/L to 3.5 × $10^9$/L, neutrophils 1.6 × $10^9$/L to 5.9 × $10^9$/L, monocytes 0.2 × $10^9$/L to 0.8 × $10^9$/L.

Supplementary Table 3). CD25$^+$ Foxp3$^+$ CXCR5$^+$ T-follicular regulatory (Tfr) cells were unlikely to contribute significantly to frequencies of CD25 expressing cTfh cells after stimulation with viral antigen since Tfr cell frequencies were very low or undetectable in the majority of samples tested (Supplementary Fig. 3). We found that the frequencies of both spike-specific and RBD-specific cTfh cells were significantly higher in severe and moderate patients than that in mild patients during acute infection (Fig. 3b). This pattern held true also during convalescence; both spike-specific and RBD-specific cTfh cells were detectable in blood of former COVID-19 patients, albeit at lower levels than during acute illness (Fig. 3b), consistent with previous studies[24–26]. Again, individuals who recovered from severe COVID-19 displayed higher frequencies of spike and RBD-specific cTfh cells even at 8 months of convalescence compared to patients who had recovered from mild disease (Fig. 3b). Frequencies of spike and RBD-specific cTfh cells positively correlated with frequencies of activated cTfh cells during acute infection (Supplementary Fig. 4), indicating that the frequency of activated cTfh cells may reflect ongoing immune reactions against the antigen or virus. During acute disease, the frequencies of spike-specific and RBD-specific cTfh cells positively correlated with both plasma antibody levels to the same viral proteins (Fig. 3c) as well as to the blood plasmablast frequency (Fig. 3d). Cytokines produced by PBMCs in response to spike and RBD were measured by Luminex, including IL-21, IFNγ, IL-4, and IL-17 (Fig. 3e). Furthermore, PBMCs isolated from severe COVID-19 patients at the acute phase of disease produced higher levels of IL-21, IFNγ, IL-4, and IL-17 in response to spike and RBD, than PBMCs from mild patients (Fig. 3e). We also found the level of plasma CXCL13 from severe COVID-19 patients was significantly higher than mild patients during acute infection (Supplementary Fig. 5). In addition, we found that the frequencies of activated and virus-specific cTfh cells in COVID-19 patients with acute disease were positively correlated with clinical parameters including age, CCI and peak disease severity (Fig. 3f). These data suggest that severe COVID-19 patients display a more robust cTfh cell response with higher level of activation markers expression and cytokines production than mild COVID-19 patients in response to SARS-CoV-2 protein, which was associated with higher frequencies of blood antibody-secreting cells and antibody titers against SARS-CoV-2 virus in severe COVID-19 patients compared to mild patients. Plasma cytokines and chemokines of COVID-19 patients with acute disease were analyzed using a proximity extension assay (Olink Proteomics) (Fig. 3g). Furthermore, we found that that frequencies of activated cTfh especially cTfh1 and cTfh2 cells, and frequencies of virus-specific cTfh cells were positively correlated with acute disease plasma levels of TNF protein family members including TNFRSF9, TNFβ, TRAIL and TWEAK, but negatively correlated with IL-6 (Fig. 3g). Our data suggest that cytokines responses induced by SARS-CoV-2 might affect generation of functional virus-specific cTfh cells against COVID-19.

**cTfh cells isolated from severe patients support plasmablast differentiation and antibody production in vitro more effectively than that from mild patients**

To confirm the functional capacity of cTfh cells from COVID-19 patients to help B-cell differentiation and antibody production, and compare the functional cTfh cell capacity in severe and mild COVID-19 patients, we isolated cTfh and non-cTfh (CD3$^+$ CD4$^+$ CD45RA$^-$ CXCR5$^-$) cells from four severe and mild COVID-19 patient samples from the acute stage of disease as well as from 4 healthy controls (Supplementary Tables 4 and 5). cTfh and non-cTfh cells were co-cultured with autologous memory or naive B cells, respectively, at a T-cell: B-cell ratio of 1:1, in the presence of Staphylococcal enterotoxin B (SEB) to enhance the T-B-cell interaction (Supplementary Fig. 6 and Fig. 4a). After 6 days co-culture of memory B cells and after 9 days co-culture of naive B cells with either cTfh or non-cTfh cells, the number of live B cells (Fig. 4b, c) as well as the frequency of differentiated CD38$^+$ CD27$^+$ plasmablasts (Fig. 4d) in the co-cultures was determined by flow cytometry. cTfh cells, but not non-cTfh cells, efficiently supported both memory and naive B-cell survival and differentiation into plasmablasts (Fig. 4b, c). Importantly, cTfh cells isolated from severe COVID-19 patients more efficiently supported memory and naive B-cell survival as well as plasmablast differentiation, as compared to cTfh cells isolated from mild COVID-19 patients or from healthy controls (Fig. 4c, d). Next, the concentration of total IgA and IgG in the co-culture supernatants was measured by ELISA. (Fig. 4e) IgA and (Fig. 4f) IgG were detectable in cTfh co-cultures with both memory B cells and naive B cells and immunoglobulin concentrations were significantly higher in co-cultures with cells isolated from severe COVID-19 patients compared to mild patients or healthy controls. In contrast, IgA or IgG were not detectable in supernatants from non-cTfh cell co-cultures (Fig. 4e, f). In line with cTfh cell functionality, IL-21 was only detectable in cTfh co-cultures (Fig. 4g). Again, IL-21 supernatant concentrations were higher in co-cultures with cells isolated from severe COVID-19 patients compared to mild patients or healthy controls (Fig. 4g), suggesting that the functionality of overall cTfh cells is more potent in severe compared to mild COVID-19.

**Appearance of SARS-CoV-2-specific cTfh cells is delayed in severe COVID-19 patients compared to patients with mild and moderate disease**

It is counter-intuitive that individuals with the highest frequencies and most functional cTfh cells as well as the highest antibody levels directed to the virus are the most severely ill patients. To resolve this paradox, we next took advantage of the fact that we had longitudinal samples during acute disease after less than two weeks of symptom onset, 2–3 weeks, 3–4 weeks as well as more than 4 weeks after symptom onset in the different disease severity groups (Supplementary Figs. 7 and 8). We found that very early after symptom onset (<2 weeks), in individuals who

**Table 2 | Demographic and clinical data of COVID-19 patients across peak disease severity**

| Peak disease severity | Mild | Moderate | Severe | Significance[a] |
|---|---|---|---|---|
| n (%) | 13 (27) | 14 (28) | 22 (45) | |
| Age in year, median (range) | 51 (29–72) | 54 (26–76) | 62 (32–77) | 0.0205 |
| Male, n (%) | 5 (38) | 9 (64) | 19 (86) | 0.0135 |
| Female, n (%) | 8 (62) | 5 (36) | 3 (14) | |
| Onset to admission[b,d], median (IQR) | NA[c] | 10 (5) | 11 (8) | 0.4438 |
| Cortisone during sample period, n (%) | 0 (0) | 2 (14) | 1 (5) | 0.5789 |
| Length of stay[b,d], median (IQR) | NA[c] | 13 (6) | 21 (12) | 0.0080 |
| **Comorbidities[e]** | | | | |
| CCI, median (IQR) | 0 (2) | 1 (3) | 2 (2) | 0.1642 |
| BMI, median (IQR) | 25.1 (6.0) | 30.1 (7.0) | 29.1 (4.7) | 0.0075 |
| Hypertension, n (%) | 1 (9) | 5 (38) | 7 (35) | 0.3173 |
| Diabetes, n (%) | 2 (18) | 6 (38) | 6 (30) | 0.5925 |
| Current smoker, n (%) | 0 (0) | 2 (13) | 0 (0) | 0.1317 |
| **Laboratory analyses in acute disease[e,f]** | | | | |
| CRP[g], median (IQR) | 1 (60) | 163 (145) | 239 (173) | <0.0001 |
| WBC[h], median (IQR) | 4.1 (1.0) | 7.8 (2.8) | 9.4 (4.3) | 0.0001 |
| Neutrophils[h], median (IQR) | 2.0 (1.1) | 6.3 (2.7) | 7.1 (4.4) | 0.0001 |
| Lymphocytes[h], median (IQR) | 1.5 (0.4) | 0.9 (0.4) | 0.7 (0.3) | 0.0084 |
| NLR, median (IQR) | 1.2 (0.2) | 6.6 (5.9) | 10.0 (9.4) | 0.0002 |
| Ct value, median (IQR) | 27.5 (9.4) | 26.3 (7.4) | 25.8 (9.1) | 0.9450 |
| **Laboratory analyses in 3-month convalescence[e,f]** | | | | |
| CRP[g], median (IQR) | 8 (21) | 14 (36) | 17 (43) | 0.0237 |
| WBC[h], median (IQR) | 1 (1) | 2 (2) | 2 (5) | 0.0610 |
| Neutrophils[h], median (IQR) | 4.8 (2.4) | 6.5 (1.9) | 6.6 (2.8) | 0.0395 |
| Lymphocytes[h], median (IQR) | 2.6 (1.4) | 4.0 (1.5) | 3.5 (1.6) | 0.5232 |
| NLR, median (IQR) | 1.7 (0.6) | 2.0 (1.0) | 1.9 (1.0) | 0.4958 |
| **Laboratory analyses in 8-month convalescence[e,f]** | | | | |
| CRP[g], median (IQR) | 9 (26) | 13 (37) | 13 (37) | 0.3000 |
| WBC[h], median (IQR) | 1 (1) | 1 (2) | 2 (2) | 0.1370 |
| Neutrophils[h], median (IQR) | 5.3 (1.1) | 7.1 (3.4) | 5.4 (3.0) | 0.0252 |
| Lymphocytes[h], median (IQR) | 3.0 (1.2) | 3.8 (1.0) | 3.2 (1.9) | 0.8488 |
| NLR, median (IQR) | 1.9 (0.7) | 2.1 (0.9) | 1.8 (1.2) | 0.3530 |

*CCI* Charlson Comorbidity Index, *NLR* neutrophil/lymphocyte ratio, *NA* data not available.

[a]Two-sided Kruskal–Wallis' and Fischer-exact tests were performed to determine statistical significance.

[b]Days.

[c]Two patients with mild disease admitted: onset to admission 14 and 8 days, length of stay 2 and 7 days, respectively.

[d]Mann–Whitney *U* was performed to determine statistical significance.

[e]Comorbidities and laboratory analyses data were available from 41 patients.

[f]Peak values: CRP. Nadir values: lymphocyte count, Ct value. WBC and neutrophil counts at the time point of the lowest lymphocyte count.

[g]mg/L.

[h]$10^9$ cells/L.

Normal range: CRP < 3 mg/L, WBC 3.5 $\times 10^9$/L to 8.8 $\times 10^9$/L, lymphocytes 1.1 $\times 10^9$/L to 3.5 $\times 10^9$/L, neutrophils 1.6 $\times 10^9$/L to 5.9 $\times 10^9$/L, monocytes 0.2 $\times 10^9$/L to 0.8 $\times 10^9$/L.

already displayed or later developed severe disease, not only the frequencies of total cTfh cells (Fig. 5a), but also the frequency of activated cTfh cells (Fig. 5b) and importantly the frequencies of spike-specific and RBD-specific cTfh cells (Fig. 5c) were significantly lower than in individuals who maintained/developed mild or moderate COVID-19. As the disease progressed, severe patients presented with increased frequencies of activated cTfh cells (Fig. 5b) and SARS-CoV-2 spike-specific and RBD-specific (Fig. 5c) cTfh cells to similar or higher levels than observed in mild and moderate patients after 4 weeks after symptom onset. These data indicate that the appearance of activated and SARS-CoV-2-specific cTfh cells was delayed in severe patients compared to moderate and mild patients during acute disease, potentially pointing to impaired generation of virus-specific cTfh cells early during infection. However, there was no significant difference in frequencies of SARS-CoV-2 spike-specific and RBD-specific CXCR5‾ memory CD4 T cells (non-cTfh cells) between severe and mild COVID-19 patients during early

infection (Fig. 5d), suggesting that the delayed appearance of virus-specific memory CD4 T cells was restricted to the cTfh cell compartment. During early infection less than 2 weeks after symptom onset, frequencies of activated non-cTfh cells were even higher in severe patients than in individuals who maintained/developed mild or moderate COVID-19 (Fig. 5e). We also found that the severe COVID-19 patients with delayed generation of activated and virus-specific cTfh cells, showed no significant difference in age, gender or CCI, but higher BMI compared with mild and moderate patients (Table 3).

**Delayed appearance of virus-specific cTfh cells is associated with delayed generation of high avidity and neutralizing plasma SARS-CoV-2 antibodies**

To understand the potential functional consequences of delayed appearance of activated and virus-specific cTfh cells, we assessed the avidity and neutralizing capacity of plasma antibodies against SARS-

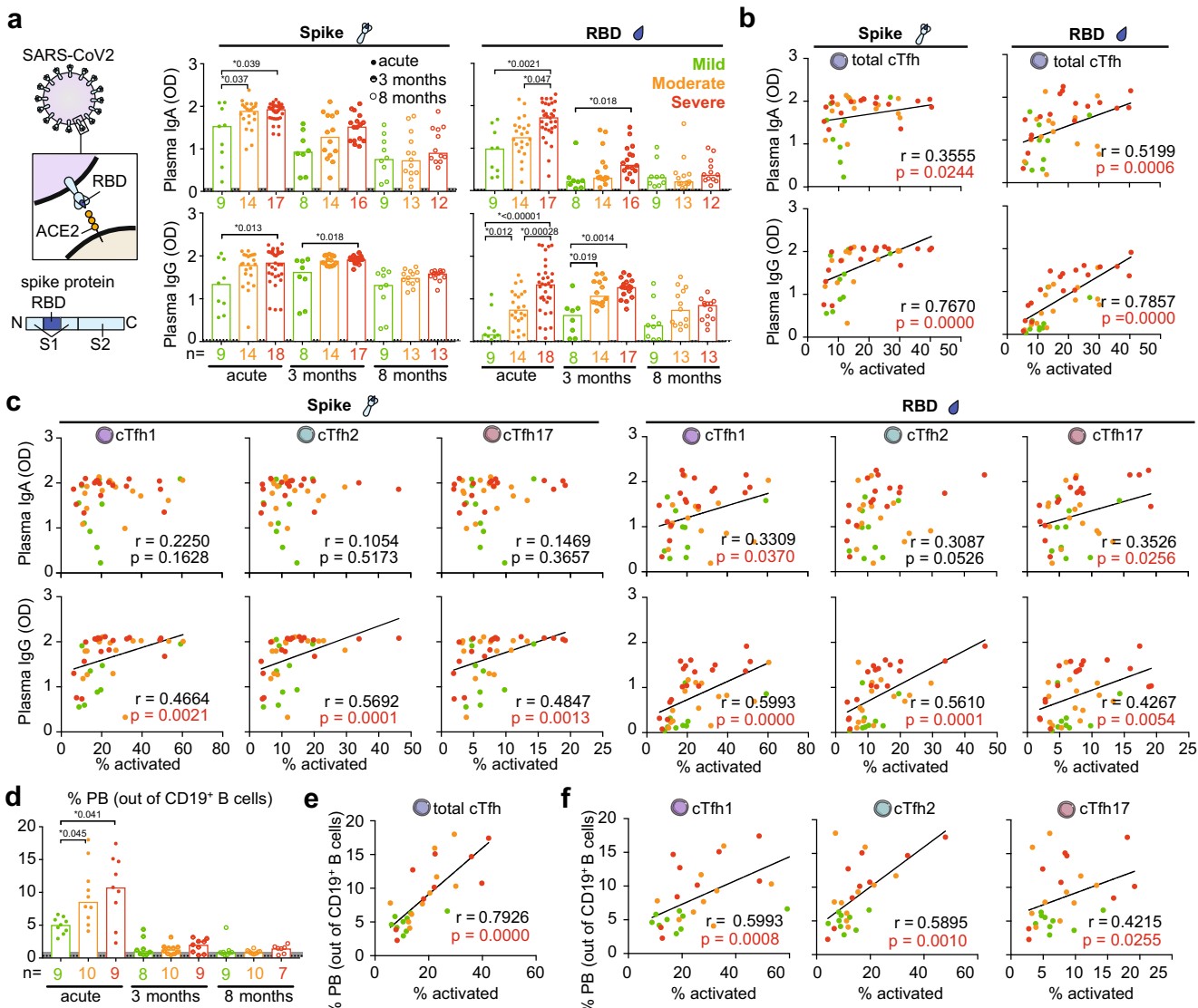

**Fig. 2 | Longitudinal titers of plasma immunoglobulins against SARS-CoV-2 spike and RBD and frequency of blood plasmablast (PB) in COVID-19 patients across disease severity from acute disease up to 8 months convalescence. a** Bar charts show the OD value of plasma IgA and IgG against SARS-CoV-2 spike and RBD with median. During acute infection (full circle), 9, 22, and 33 individual samples from 9 mild, 14 moderate, and 18 severe patients, respectively, were analyzed using two-sided Generalized Estimating Equations (GEE) to account for the intra-person correlations inherent to repeated measures and assess statistically significant differences without adjusting for multiple comparisons. During convalescence (half and open circle), only one sample from each patient was analyzed and two-sided Kruskal–Wallis with Dunn's multiple comparisons test was used to assess statistically significant differences. One severe patient displayed IgA deficiency over time was excluded in all IgA analyses. **b**, **c** Two-sided Spearman correlation for plasma immunoglobulins against the spike and RBD versus frequency of activated (**b**) cTfh and **c** cTfh subsets during acute disease. In all, 9, 14, and 18 individual samples from

9 mild, 14 moderate and 18 severe patients were analyzed. For patients with longitudinal acute samples, data from the earliest sample was involved as the representative in Spearman correlation analysis. **d** Frequency of plasmablast from COVID-19 patients is shown by bar chats with median, and the dotted lines show the median frequency with 95% CI (gray area) of healthy controls. Dots are individual samples color-coded according to peak disease severity. *X* axis shows the number of patients in each bar. Only one sample from each patient was analyzed during both acute disease and convalescence. Two-sided Kruskal–Wallis with Dunn's multiple comparisons test was used to consider all statistically significant. **e**, **f** Two-sided Spearman correlation for frequency of plasmablast versus frequency of (**e**) activated cTfh and **f** activated cTfh subsets during acute disease. Overall, 9, 10, and 9 individual samples from 9 mild, 10 moderate and 9 severe patients were analyzed during acute disease. $P < 0.05$ was considered to be a significant difference. *$P$ values <0.05 are listed above each comparison. Source data are provided as a Source Data file.

CoV-2 from COVID-19 patients during acute disease. In line with the cTfh data, we found that in individuals who already displayed or later developed severe disease, the avidity index of plasma IgG antibodies against spike protein (Fig. 5f and Supplementary Fig. 9a) and the plasma neutralization $IC_{100}$ against SARS-CoV-2 (isolate SARS-CoV-2/human/SWE/01/2020) (Fig. 5h) were significantly lower than in patients who maintained or developed mild or moderate COVID-19 within 2 weeks after symptom onset. However, the antibody avidity index and neutralizing capacity increased as disease progressed and

reached similar or higher levels than moderate patients after 4 weeks of symptom onset (Fig. 5f, h and Supplementary Fig. 9b). The frequency of activated and SARS-CoV-2 spike-specific cTfh cells correlated strongly with the avidity index of plasma IgG antibodies (Fig. 5g) and plasma neutralization $IC_{100}$ (Fig. 5i). Though the titers of plasma IgA and IgG antibodies against SARS-CoV-2 spike and RBD in severe patients showed similar or even higher level compared with mild COVID-19 patients during early infection (Supplementary Fig. 10a, b), the plasma neutralization potency (calculated as $IC_{100}$ divided by

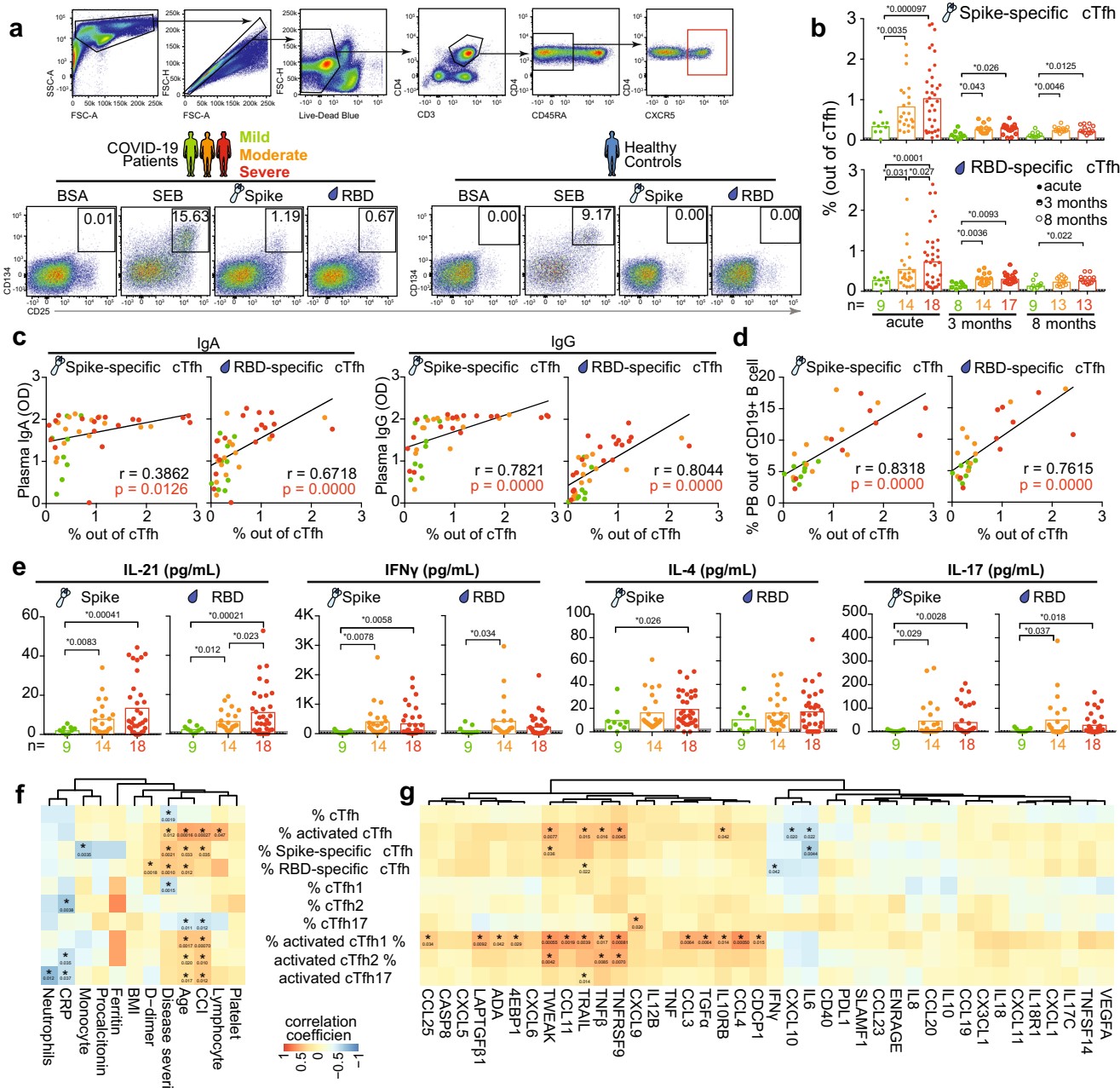

**Fig. 3 | Longitudinal frequency of SARS-CoV-2 spike and RBD-specific cTfh in COVID-19 patients across disease severity from acute disease up to 8 months convalescence. a** Representative example with gating strategy to identify SARS-CoV-2 spike and RBD-specific cTfh. **b** Bar charts show the median frequency of SARS-CoV-2 spike and RBD-specific cTfh from acute disease (full circle) to convalescence (half and open circle) with median. Dots are individual samples color-coded according to peak disease severity. Dotted lines show the median frequency with 95% CI (gray area) of healthy controls. During acute disease, 9, 22 and 33 individual samples from 9 mild, 14 moderate and 18 severe patients, respectively, were analyzed using two-sided Generalized Estimating Equations (GEE) to account for the intra-person correlations inherent to repeated measures and assess statistically significant differences without adjusting for multiple comparisons. During convalescence, only one sample from each patient was analyzed using two-sided Kruskal–Wallis with Dunn's multiple comparisons test to assess statistically significant differences. **c**–**d** Two-sided Spearman correlation for frequency of virus-specific cTfh versus (**c**) titers of plasma immunoglobulins against the spike and

RBD, and versus (**d**) frequency of plasmablast during acute disease. For patients with longitudinal acute samples, data from the earliest sample was involved as representative in Spearman correlation analysis. **e** Bar charts show the median concentration of cytokines in supernatants of spike and RBD protein-stimulated PBMCs from COVID-19 patients with acute disease. Dotted lines show the median concentration of cytokines with 95% CI (gray area) in supernatants from healthy controls. Two-sided GEE was used to account for the intra-person correlations inherent to repeated measures and assess statistically significant differences without adjusting for multiple comparisons. **f**, **g** Heatmap summarizing the inter-relationship between characteristics of cTfh cells and **f** clinical parameters, and **g** levels of plasma cytokines/chemokines from COVID-19 patients with acute disease. Two-sided repeated measures correlations without multiple comparisons were calculated. **e**–**g** In all, 9, 22, and 33 individual samples from 9 mild, 14 moderate and 18 severe patients, respectively, were analyzed. $P < 0.05$ was considered to be a significant difference. *P values <0.05 are listed above each comparison and in the heatmaps. Source data are provided as a Source Data file.

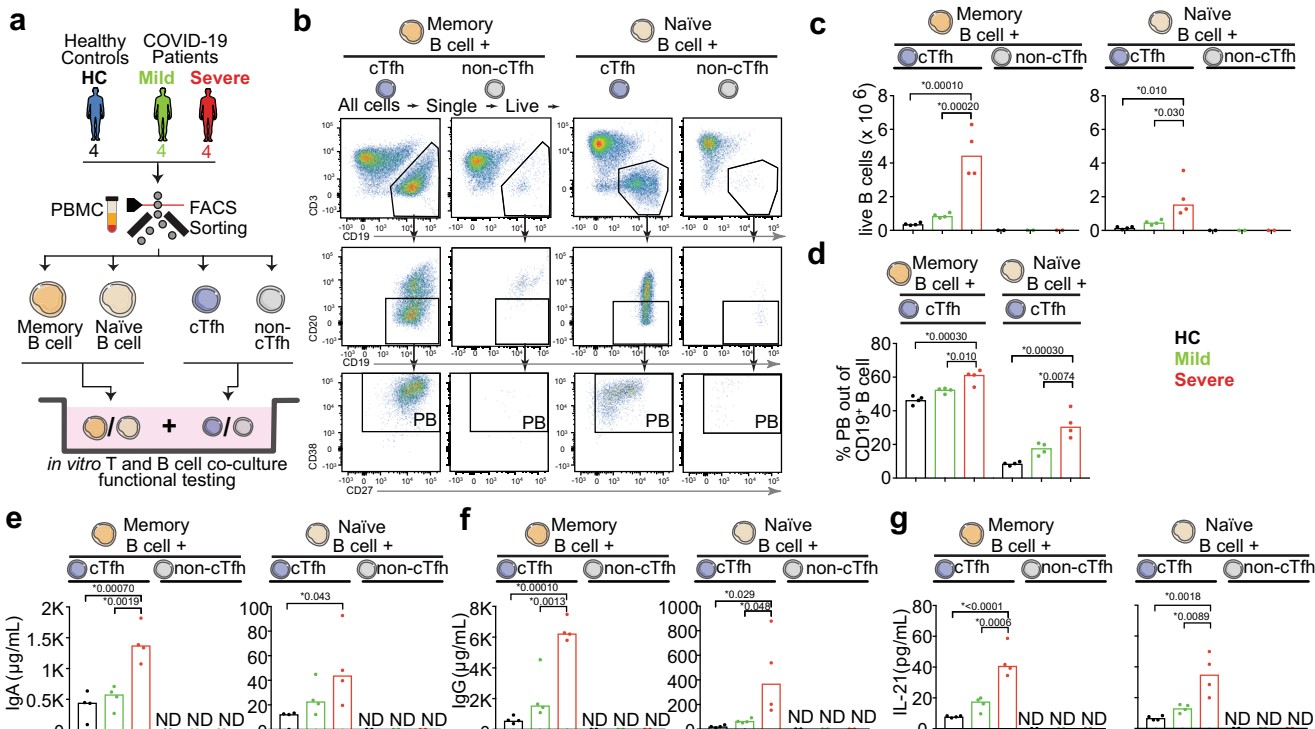

**Fig. 4 | cTfh isolated from severe COVID-19 patients with acute disease support B cells differentiation and antibody production more efficiently than that from mild COVID-19 patients. a** Blood cTfh, non-cTfh, memory B, and naive B cells were isolated from four severe and mild COVID-19 patients with acute disease, as well as four healthy donors. cTfh or non-cTfh cells were cultured with autologous memory B cells for 6 days, or with autologous naive B cells for 9 days respectively. **b** Representative example with gating strategy to identify (left) memory B-cell and (right) naive B-cell differentiation. **c** Bar charts show the number of live B cells in (left) cTfh/non-cTfh co-cultured with memory B cells and (right) cTfh/non-cTfh co- cultured with naive B cells. **d** Bar charts show the frequency of plasmablast in cTfh-memory/naive B-cell co-cultured. **e, f** Bar charts show the concentration of (**e**) IgA and **f** IgG in supernatant from cTfh/non-cTfh co-cultured with (left) memory B cells and (right) naive B cells. **g** Bar charts show the concentration of IL-21 in supernatant from cTfh/non-cTfh co-cultured with (left) memory B cells and (right) naive B cells. **e–g** ND means not detectable. Differences were tested with One-Way ANOVA. $P < 0.05$ was considered to be a significant difference. *$P$ values <0.05 are listed above each comparison. Source data are provided as a Source Data file.

antibody titers) in severe patients was significantly lower than mild and moderate patients (Fig. 5j). As failure to achieve high avidity and neutralizing antibodies has been shown to result in a lack of protective immunity against SARS-CoV-2 infection[34], our study suggests that impaired generation of functional virus-specific cTfh cells delays the production of protective antibodies to combat the infection at an early stage and thereby enables progression to more severe COVID-19 disease. In summary, delayed generation of activated and SARS-CoV-2-specific cTfh resulted in a delayed production of high-quality antibodies for protection, which may consequently worsen COVID-19 disease progression.

## Discussion

It is still unclear what immunological parameters dictate the broad spectrum of clinical outcomes after SARS-CoV-2 infection. Though several studies have reported a positive relationship between disease severity and robust SARS-CoV-2-specific antibody or T-cell responses[24,32,35], how T-cell and antibody responses affect COVID-19 disease progress are still unclear. In this study, we focus on cTfh cells, which originated from lymph nodes[16,17] and display clonal and developmental overlap with GC Tfh cells as supported by epigenetic, transcriptomic and TCR repertoire studies[16,36]. Thus, cTfh cells are considered a circulating counterpart of bona fide lymphoid Tfh cells and a reflection of the GC reaction in response to antigen[23,37,38]. In SARS-CoV-2/COVID-19, studies on cTfh cells have focused on convalescent individuals[24–26] while longitudinal data on characteristics of cTfh cells in COVID-19 patients during acute infection and the correlation with disease severity have been missing. Whether cTfh cell

functionality is impaired by viral infection, and whether this connects with disease severity remains unclear. Here, we investigated the characteristics and function of cTfh cells in COVID-19 patients with mild to severe disease longitudinally during acute infection and convalescence in a clinically well-characterized cohort.

In this study, we observed a delayed appearance of activated and SARS-CoV-2-specific cTfh but not non-cTfh cells in severe compared to mild COVID-19 patients during early infection (less than 2 weeks after symptom onset). Delayed generation of high-avidity and neutralizing antibodies against SARS-CoV-2 were also observed in severe compared to mild COVID-19 patients during the same period of early infection, which strongly correlated with frequencies of both activated and virus-specific cTfh cells, indicating that the delayed generation of activated and virus-specific cTfh cells was associated with a delayed production of high quality of productive antibodies. This may lead to reduced capacity to prevent viral spread at early stages of infection and consequently may enable progression to more severe COVID-19 disease. The delayed generation of functional virus-specific cTfh cells in severe patients observed in our study may reflect the delayed germinal center reaction in severe compared with mild patients, which supports and potentially explains the observation from other labs that patients with mild COVID-19 displayed substantial affinity maturation for antibodies binding to the prefusion spike much earlier than patients with severe COVID-19[39], and blunted affinity maturation against the SARS-CoV-2 prefusion spike protein may predict worse outcome for hospitalized patients[40]. We also found that at early infection, severe COVID-19 patients who had a lower frequency of functional cTfh cells displayed similar or even higher plasma antibody titers compared with mild

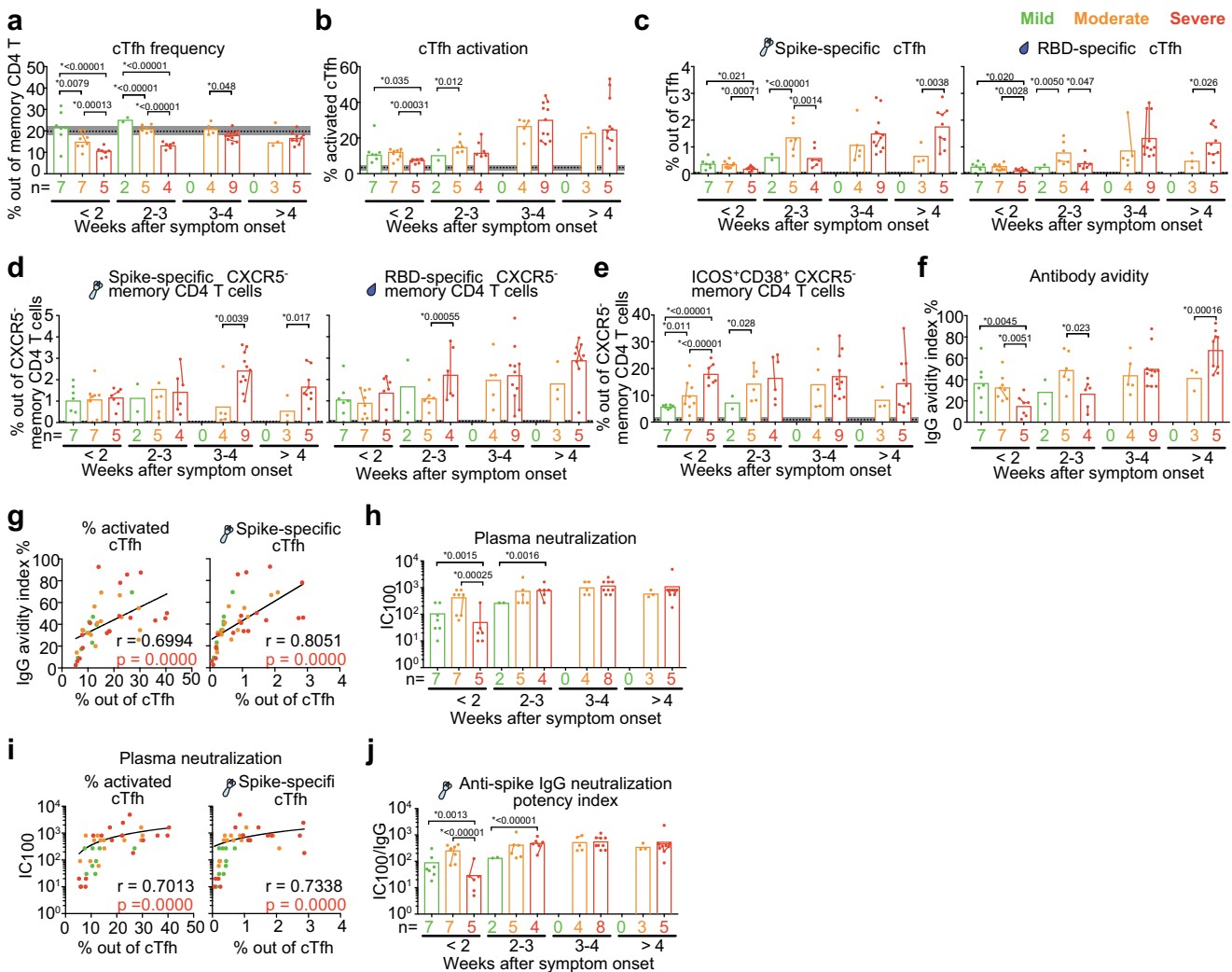

**Fig. 5 | Delayed emergence of activated and virus-specific cTfh cells correlated with delayed emergence of high avidity and neutralization plasma antibodies in severe COVID-19 patients compared to mild and moderate patients. a–e** Bar charts show the mean frequency of (**a**) cTfh cells, (**b**) activated cTfh cells, SARS-CoV-2 spike, and RBD-specific (**c**) cTfh and (**d**) CXCR5⁻ memory CD4 T cells, (**e**) activated CXCR5⁻ memory CD4 T cells in patients with acute COVID-19 according to weeks after symptom onset. **f** Bar charts show the avidity index of plasma IgG during acute disease against SARS-CoV-2 spike over weeks after symptom onset with mean. **g** Two-sided Spearman correlation for frequency of activated and virus-specific cTfh versus avidity index of plasma IgG during acute disease. **a–g** In total, 9, 22, and 33 individual samples from 9 mild, 14 moderate, and 18 severe patients, respectively, were analyzed. **h, j** Bar charts show the (**h**) neutralization IC₁₀₀ and **j** anti-spike IgG neutralization potency index of COVID-19 patient plasma samples across peak disease severity over weeks after symptom onset. **i** Two-sided

Spearman correlation for frequency of activated and virus-specific cTfh versus plasma neutralization IC100 during acute disease. **h–j** Overall, 9, 22, and 31 individual samples from 9 mild, 14 moderate, and 17 severe patients, respectively, were analyzed. Dots are individual samples color-coded according to peak disease severity. The dotted lines show the mean frequency with 95% CI (gray area) of healthy controls. X axis shows number of patients in each bar. Dots from same patient are linked with line in each bar. The graphical presentations of the different outcomes in (**a–f, h, j**) were based on a two-sided Generalized Estimating Equations model with time modeled using a restricted cubic spline with knots at 0, 14, 21, 28, and 53 days, without adjusting for multiple comparisons. **g, i** One individual sample from each patient was analyzed for Two-sided Spearman correlation during acute disease. For patients with longitudinal acute samples, data from the earliest sample was involved. P < 0.05 was considered to be a significant difference. *P values <0.05 are listed above each comparison. Source data are provided as a Source Data file.

patients, indicating the possibility of extrafollicular antibody responses. However, these early Tfh-independent antibody responses in severe patients were much lower in avidity and neutralizing capacity and consequently less efficient in controlling viral spread which might contribute to disease progression, compared with mild and moderate patients who displayed higher frequencies of functional virus-specific cTfh cells during early infection.

The immunological reason for the impaired generation of cTfh cells could be a result of complicated dysfunction of immune responses such as lymphopenia, aberrant cytokine/chemokine production and dysregulated innate immune response in COVID-19 patients with severe disease. Unfortunately, in the majority of our severe and moderate patients, the count of lymphocytes was not

tested at the same day when the samples were collected for this study, thus we could not accurately assess the absolute number of cTfh cells in blood or measure the impact of lymphopenia on cTfh cells, which is a limitation of this study. But our Olink proteomics data show frequencies of activated and virus-specific cTfh cells negatively correlated with IL-6 (Fig. 3g). IL-6 has been considered a signal of a dysregulated cytokine response contribute to severe COVID-19 outcome[41], suggesting the aberrant cytokine response might be associated with impaired generation of cTfh cells. The Olink proteomics data also suggest TNF family members, especially TNFRSF9, TNFβ, TRAIL and TWEAK, positively correlate with activation of cTfh cells and generation of virus-specific cTfh cells (Fig. 3g). Interestingly, during the same period of early infection when the generation of functional

**Table 3 | Demographic and clinical data of COVID-19 patients sampled less than two weeks after symptoms onset**

| Peak disease severity | Mild | Moderate | Severe | Significance[a] |
|---|---|---|---|---|
| n (%) | 7 (37) | 7 (37) | 5 (24) | |
| Age in year, median (range) | 38 (29–72) | 50 (41–65) | 49 (32–62) | 0.0775 |
| Male, n (%) | 2 (14) | 4 (56) | 4 (90) | 0.6867 |
| Female, n (%) | 5 (86) | 3 (44) | 1 (10) | |
| **Comorbidities** | | | | |
| CCI, median (IQR) | 0 (0) | 1 (3) | 1 (2) | 0.3571 |
| BMI, median (IQR) | 23.0 (7.7) | 30.8 (8.4) | 30.6 (12.5) | 0.0273 |
| Hypertension, n (%) | 0 (0) | 1 (14) | 0 (0) | 0.4046 |
| Diabetes, n (%) | 1 (14) | 2 (28) | 1 (20) | 0.8048 |
| Current smoker, n (%) | 0 (0) | 2 (33) | 0 (0) | 0.1472 |

CCI Charlson Comorbidity Index.
[a]Two-sided Kruskal–Wallis' and Fisher's exact tests were performed to determine statistical significance.

virus-specific cTfh cells was impaired, relatively lower levels of TNFβ and TWEAK were also observed in patients who had displayed or later developed severe COVID-19, compared with mild and moderate patients, though differences were not statistically significant due to the limited sample size (Supplementary Fig. 11). Studies have shown that the engagement of ICOS on T cells by ICOS ligand (ICOS-L) on B cells is crucial for Tfh cell generation[42]. Noncanonical NF-κB activation is considered an important factor for maintaining high levels of ICOS-L expression in B cells, and consequently important for antigen-stimulated Tfh cell differentiation[43]. While both TNFβ and TWEAK have been reported to induce the noncanonical NF-κB pathway[44], we could argue that low levels of TNFβ and TWEAK might be associated with impaired virus-specific cTfh cell generation and activation in severe patients during early infection. This is supported by another study showing that decreased expression of the costimulatory molecule ICOS-L on B cells might impair Tfh cell activity in hospitalized compared with ambulatory COVID-19 patients[45]. Altogether, our study provides some potential clues that aberrant chemokine production might contribute to the delayed virus-specific cTfh cell generation in severe patients during early infection period.

In addition, we found that characteristics of cTfh cells are associated with disease severity in COVID-19 patients during acute infection. Our data show that severe COVID-19 patients displayed the lowest frequency of total cTfh, cTfh1 and cTfh17 cells, in line with previous reports[46,47]. However, activated ICOS+CD38+ cTfh cells dramatically increase in moderate and severe COVID-19 patients compared with mild patients during acute disease, suggesting that activated cTfh cells may reflect recent or ongoing virus encounter and emigration of cTfh cells from the GCs, similar to observations from others[48,49]. This is in line with several studies trying to understand the immune profile of COVID-19 patients[4,28,48] that report elevated frequencies of CD38+ ICOS+ CXCR5+ CD4+ T cells during acute disease. Importantly, we also found that the frequencies of SARS-CoV-2 spike-specific and RBD-specific cTfh cells were higher in severe compared to mild patients. This was further supported by cytokine data, showing that higher levels of typical cTfh cytokines were secreted from PBMCs in response to SARS-CoV-2 spike and RBD proteins from severe COVID-19 patients compared to mild patients. It should be noted that in the current study, virus-specific cTfh cells were identified using an activation-induced marker (AIM) assay. Importantly, a study by Nelson and colleagues utilized a peptide:MHCII tetramer-based strategy to identify virus-specific cTfh cells and also observed that SARS-CoV-2-specific cTfh cells persisted several months after recovering from the acute infection[50], which is in line with the observation in our study. This suggests that there is long-term maintenance of SARS-CoV-2-specific cTfh cells after COVID-19. However, Nelson and colleagues observed that a higher proportion of the virus-specific CD4 T cells displayed a higher proportion a cTfh cell phenotype (CXCR5+) in convalescent samples from previously non-hospitalized persons compared to samples from previously hospitalized persons. In contrast, we observed that cTfh cells from individuals who recovered from mild COVID-19 displayed a lower proportion of virus-specific cells compared to individuals who recovered from severe COVID-19 during both 3 and 8 months of convalescence. These differences may relate to differences in the methods used, or time of sampling and/or definition of COVID-19 disease severity. Future studies should compare tetramer stainings with the AIM assay side by side in longitudinal sample sets across disease severity to address this. We also assessed the age, gender and comorbidities previously demonstrated to be associated with disease severity[1,30] and found that our severe patients were older and more frequently male when compared with mild and moderate patients, similar to other reports[1,30]. Interestingly, we found frequencies of activated and virus-specific cTfh cells positively correlate with age and comorbidities, indicating that, along with other clinical parameters, cTfh cells could be considered as a potential immunological parameter that correlate with disease severity.

Furthermore, to compare the functionality of cTfh cells across COVID-19 disease severity, we performed autologous cTfh-B-cell cocultures in vitro. We found that cTfh cells but not non-cTfh cells from COVID-19 patients supported plasmablast differentiation and antibody generation in vitro, and that the functionality of overall cTfh cells from severe COVID-19 patients was even more potent than cTfh cells from mild patients. One possible reason for this might be that cTfh cells isolated from severe COVID-19 patients produce higher amounts of IL-21, a critical cytokine for B-cell differentiation and antibody production, compared to cTfh cells isolated from mild patients. Blocking IL-21 would inhibit the generation of plasma cells in vitro[23]. Another potential explanation for the superiority of cTfh cells from severe COVID-19 patients is that the frequency of activated ICOS+ cTfh cells is higher in severe compared to mild patients (Supplementary Fig. 12). ICOS, a member of the CD28 family of T-cell co-stimulators upregulated in activated T, facilitate T-B-cell interaction by binding to its ligand (ICOS-L) expressed on B cells[51,52]. ICOS+ cTfh cells have been reported to be more efficient in supporting B-cell differentiation and antibody production than ICOS− cTfh cells in vitro[23]. The difference in sampling time after symptom onset of mild and severe COVID-19 patients for samples used for functional assays (Supplementary Table 5) likely contributed to a difference in frequencies of activated cTfh cells. Unfortunately, we did not compare the priming of naive and memory B cells isolated from COVID-19 patients, which likely contributed to a difference in frequencies of differentiated plasmablast, which is a limitation of this study. Due to the limited blood volume we were able to obtain from each COVID-19 patient, it was not feasible to isolate sufficient numbers of activated vs. non-activated cTfh cells, or SARS-CoV-2-specific vs. non-specific cTfh cells for subsequent coculture with autologous B cells primed the same way, which is also a

limitation of this study. Another limitation of this study is that virus-specific cTfh cells function was not tested by adding viral proteins instead of superantigen partly due to the limited cell numbers, as well as published data from the Ueno lab showing that naive B cells co-cultured with CXCR5[+]CD4[+] T cells did not produce antibodies in the absence of SEB[22].

In summary, we provide functional evidence that cTfh cells from COVID-19 patients help antibody-secreting B-cell differentiation and antibody generation, and that cTfh cell characteristics are associated with disease severity. Generation of activated and SARS-CoV-2-specific cTfh is delayed early during infection in patients who proceed to develop severe COVID-19, resulting in a delayed production of high-quality antibodies for protection that in turn might worsen disease progression. Our study provides helps to understand the potential immunological factors associated with disease severity in COVID-19 and suggests that identifying agents that could restore impaired cTfh cell generation may be of therapeutic value in the future.

## Methods

### Study approval
The study was approved by the Swedish Ethical Review Authority, and performed according to the Declaration of Helsinki. Written informed consent was obtained from all patients and controls. For sedated patients, the denoted primary contact was contacted and asked about the presumed will of the patient and to give initial oral and subsequently signed written consent. When applicable, retrospective written consent was obtained from patients with non-fatal outcomes.

### Study design and patient inclusion
COVID-19 patients confirmed by SARS-CoV-2 PCR test were enrolled at the Karolinska University Hospital and Haga Outpatient Clinic (Närakut Haga), Stockholm, Sweden. Additionally, patients with mild disease were included by recruiting household contacts who were SARS-CoV-2 PCR positive. Gender and days after symptom onset were determined based on self-report. In total 147 COVID-19 adult patients were recruited as previously reported[33,53]. For this study, we selected both male and female patients with non-fatal outcome and available biobanked longitudinal acute and convalescent samples, and excluded patients with autoimmune disease or hematological malignancies. Consequently, 41 COVID-19 patients were included in this study. These patients were subsequently sampled at ~3 ($n = 39$ patients) and 8 ($n = 35$ patients) months after recovery. Blood samples from 20 age and gender-matched healthy controls (HCs) recruited prior to the pandemic were also included (Fig. 1a). Clinical characteristics were compared between the COVID-19 patient cohort and HCs (Table 1). The total burden of comorbidities was assessed using the Charlson Comorbidity index (CCI)[54]. Disease severity was assessed daily in admitted patients, according to the respiratory domain of the sequential organ failure assessment (SOFA) score[55], with additional levels for non-admitted patients as mild cases (Fig. 1b). Patients were grouped depending on their peak disease severity. Medical records from grouped patients across disease severity were analyzed (Table 2). We included the vast majority of our study participants (9 mild, 14 moderate and 18 severe) in March-May 2020, when corticosteroid use was still not common practice in Stockholm, Sweden. Therefore, only 3 COVID-19 patients were treated with cortisone (Table 2), and one was sampled before cortisone was administered. For functional co-culture experiments, we recruited new patients (4 mild and 4 severe) in December 2020–March 2021 that did not receive corticosteroid treatment during the sampling period. However, two patients were sampled 26 and 37 days after corticosteroid treatment was concluded. None of the patients in this study received IL-6 inhibitors (tocilizumab) or IL-1 inhibitors during the study/sampling period.

### Flow cytometry
Staining for subsets of cTfh cells and plasmablast in PBMCs was performed on cryopreserved samples based on our earlier protocols[56,57] with modifications. Cells were stained using Live/Dead Blue (Invitrogen, cat no. L34962) or Live/Dead Aqua (Invitrogen, cat no. L34966) first and then incubated with human FcR blocking reagent (Miltenyi Biotec) and stained with appropriate combination of fluorescently labeled monoclonal antibodies in appropriate dilutions (Supplementary Tables 1–4) for 20 min at 4 °C. Cells were washed with PBS and fixed with 1% paraformaldehyde. Samples were acquired using LSRFortessa flow cytometer (BD). Data were analyzed using FlowJo version 10 (TreeStar).

### Cytokine and chemokine analysis
The concentration of cytokines in the supernatants from PBMCs stimulation assay and cTfh-B-cell co-culture assay was determined by Custom-made Luminex assay (R&D Systems, cat no. LXSAHM-05), including IL-21, IFNγ, IL-4, IL-17, and TNF based on the manufacturer's protocol. Briefly, supernatants were incubated with the microparticle cocktail on a shaker (800 rpm) for 2 h at room temperature (RT). Then wells were washed and incubated with biotin-antibody cocktail on a shaker (800 rpm) for 1 h at RT. Next wells were washed and incubated with Streptavidin-PE on a shaker (800 rpm) for 0.5 h at RT. Subsequently, wells were washed and diluted in wash buffer and the results were generated by Bio-Plex 200 analyzer (BIO-RAD).

Plasma chemokine CXCL13 was analyzed using aptamer-based SOMAscan proteomic discovery platform v4.1 (SOMAlogic Inc., Boulder, CO), according to the manufacturer's protocols[58]. The discovery platform allows the detection of ~7000 protein analytes using modified single-stranded DNA-based protein affinity reagents referred as SOMAmers (Slow Off-rate Modified Aptamers). The data generated from this assay passed through multiple standardization and normalization steps, including adaptive normalization by maximum likelihood to minimize intra- and interassay variation. The quantitative levels of protein analytes were presented as relative fluorescence unit (RFU). For this paper, only CXCL13 data were included.

Plasma cytokines and chemokines were also tested by Olink Proteomics assay (92-biomarker Inflammation panel) performed based on the manufacturer´s instructions[59]. Briefly, DNA oligonucleotides labeled antibodies combine with target antigen and oligonucleotides were hybridized and extended by DNA polymerase. Then protein expression levels were determined by high-throughput real-time PCR and presented as normalized protein expression (NPX) values. Values were calculated from inverted Ct values, with a high NPX value corresponding to a high protein concentration. Intensity normalization of data was performed according to the manufacturer's protocol, to minimize intra- and interassay variation.

### Detection of SARS-CoV-2-specific cTfh cells
Recombinant SARS-CoV-2 spike and receptor binding domain (RBD) proteins were received through the global health-vaccine accelerator platforms (GH-VAP) funded by the Bill & Melinda Gates Foundation. Cryopreserved PBMCs were thawed and rested for 2 h at the incubator. Then cells were cultured in 96-well U-bottomed plates ($0.5 \times 10^6$ cells per well) and exposed to 5 µg/mL of protein (SARS-CoV-2 spike, SARS-CoV-2 RBD and BSA), or 5 µg/mL of staphylococcal enterotoxin B (SEB) for 20 h. After stimulation, cells were collected and analyzed by flow cytometry as described above; culture supernatants were harvested to determine concentrations of cytokines by Luminex assay.

### Serology
The titers of plasma IgA and IgG binding against SARS-CoV-2 spike trimer or receptor binding domain (RBD) monomer were determined by enzyme-linked immunosorbent assay (ELISA). Recombinant SARS-CoV-2 spike and RBD proteins were received through the global health-

vaccine accelerator platforms (GH-VAP) funded by the Bill & Melinda Gates Foundation. Briefly, 96-half well plates were coated with 50 ng/well of the respective protein. Plates were incubated with duplicate dilution (1:20) of each plasma sample at ambient temperature for 2 h. Detection was performed with a goat anti-human IgG HRP-conjugated secondary antibody from BD Biosciences (Clone G18-145, cat no. 555788), or polyclonal goat anti-human IgA HRP-conjugated secondary antibody from Thermo Fisher (cat no. A18781) followed by incubation with TMB substrate (BioLegend, cat no. 421101) which was stopped with a 1 M solution of $H_2SO_4$. Absorbance was read at 450 nm + 550 nm (background correction) using an ELISA reader. Data are represented as the mean OD value of the two duplicates.

Plasma antibody avidity were tested for by an anti-SARS-CoV-2 ELISA[56,60]. Briefly, 96-half well plates were coated with 50 ng/well of the recombinant SARS-CoV-2 spike protein. Plates were incubated with fivefold dilution series of each plasma sample at ambient temperature for 2 h and then washed with 0.1% Tween-20/PBS washing buffer. To determine the relative avidity index, two rows of microplate wells next to each other were used for each donor. In one row wells were incubated with PBS and in the other row wells were incubated with 1.5 M solution of sodium thiocyanate for 10 min at ambient temperature, followed by washing with washing buffer. The sodium thiocyanate treatment leads to the detachment of low-avidity antibodies from the antigen. Detection was performed with a goat anti-human IgG HRP-conjugated secondary antibody from BD Biosciences (Clone G18-145) followed by incubation with TMB substrate (BioLegend) which was stopped with a 1 M solution of $H_2SO_4$. Absorbance was read at 450 nm + 570 nm (background correction) using an ELISA reader and the half-maximal effective concentration ($EC_{50}$) was calculated using GraphPad Prism 9. The relative avidity index was calculated from the ratio of the $EC_{50}$ with and without sodium thiocyanate treatment and is expressed as a percentage.

Neutralization of replicating SARS-CoV-2 in plasma samples from COVID-19 patients was as described previously[61,62]. Briefly, plasma from COVID-19 patients was 3-fold serially diluted, mixed with SARS-CoV-2 (isolate SARS-CoV-2/human/SWE/01/2020, https://www-ncbi-nlm-nih-gov.proxy.kib.ki.se/nuccore/MT093571), incubated for 1 h and added, in duplicates, to confluent Vero E6 cells in 96-well plates. After 5 days of incubation, wells were inspected for signs of cytopathic effects (CPE) by optical microscopy. Each well was scored as either neutralizing (if no signs of CPE were observed) or non-neutralizing (if any CPE was observed). The arithmetic means neutralization titer of the reciprocals of the highest neutralizing dilutions from the two duplicates for each sample was then calculated to determine the $IC_{100}$.

### Functional cTfh cell assay

cTfh, non-cTfh, naive, and memory B cells were sorted from frozen PBMCs from four severe and mild COVID-19 patients with acute disease, respectively, and three healthy donors, using FACS Aria Fusion (BD) with a 100 μm nozzle (Supplementary Fig. 6 and Supplementary table). Sorted cTfh or non-cTfh cells ($5 \times 10^4$ cells per well) were co-cultured with autologous memory B cells ($5 \times 10^4$ cells per well) for 6 days and with autologous naive B cells ($5 \times 10^4$ cells per well) for 9 days in complete RPMI 1640 + 10% FCS in the presence of SEB (1 μg/mL) in 96-well U-bottomed plates. Culture supernatants were harvested at day 2 to measure cytokines by Luminex assay. After 6 and 9 days of culturing, cells were harvested to determine B-cell count and differentiation by flow cytometry as described above, and culture supernatants were collected to test IgA and IgG concentrations by ELISA.

### Statistical analyses

All statistical analyses were performed using R (version 4.1.2), Prism 9 (GraphPad) and SPSS version 27.0 (IBM, New York). Grouped data are generally presented as means unless otherwise specified. The estimating of all regression models was done using two-sided Generalized Estimating Equations (GEE) to account for the intra-person correlations inherent to repeated measures. An exchangeable correlation structure was used. Adjusted estimates for average differences in the different outcomes between groups were calculated with a linear model. Due to the exploratory nature of the analysis, no adjustment for multiple comparisons was made. Time-period-specific differences between groups were calculated by using time-period-specific subsets of the data. The graphical presentations of the different outcomes were based on a GEE model with time modeled using a restricted cubic spline with knots at 0, 14, 21, 28, and 53 days. For heatmaps, repeated measures correlations were calculated, except for age, BMI, CCI, D-dimer, and ferritin, which either had no time-specific variation (age, BMI, CCI) or too few values. For these variables the correlations were calculated by collapsing the timepoints to one mean value/person. In the presence of repeated measures, parametric correlation measures based on an ANCOVA model was chosen, as there are no nonparametric correlation coefficients for repeated measures. For the heatmaps, when there were no repeated measures, Spearman rho was used. Data of all convalescent samples were performed using the Kruskal–Wallis test. For the clinical tables, data was presumed to have a non-standard distribution and comparisons between continuous variables were performed using the Mann–Whitney $U$ test or Kruskal–Wallis' test where appropriate. In the case of using Kruskal–Wallis' test, multiple comparisons were performed using Dunn's test. Nominal variables were compared between groups using Fisher's exact test. Interdependence of two non-categorical parameters was performed using Spearman's rho. A significance level of 95% was used and all tests were two-sided. In the case of using Spearman's rho when there were repeated samples, the data from the earliest sample was involved as representative for patients with longitudinal acute samples.

### Reporting summary

Further information on research design is available in the Nature Portfolio Reporting Summary linked to this article.

## Data availability

The personal data are not publicly available due to them containing information that could compromise research participant privacy. All other data are provided in the article and its Supplementary files or from the corresponding author upon request. Source data are provided with this paper.

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

## Acknowledgements

We thank the patients and healthy volunteers who have contributed to this study. We would also like to thank the hospital staff for assistance with patient sampling, collection of clinical data, and sample processing. We also thank Neil King, Lauren Carter, and members of the Institute for Protein Design Core Laboratories, University of Washington, Seattle, WA, United States for recombinant protein reagents. We also thank Tyson H. Holmes for the analysis of Olink data and revision of the manuscript. This work was supported by grants awarded to Anna Smed-Sörensen from the Swedish Research Council (2020-06100, 2020-05764, 2020-06312, 2021-03046), the Swedish Heart-Lung Foundation (20220143, 20210085, 20200034), the Bill & Melinda Gates Foundation (INV-018945), the Knut and Alice Wallenberg Foundation through SciLifeLab and Karolinska Institutet. Paulo Czarnewski is financially supported by the Knut and Alice Wallenberg Foundation (KAW 2017.0003) as part of the National Bioinformatics Infrastructure Sweden at SciLifeLab.

## Author contributions

Experimental study design: M.Y. and A.S.-S. Patient recruitment and sample collecting: S.F.-J., B.Ö., L.A., E.Å., R.F.-J., M.B., N.J., and A.F. Anonymized patient clinical data collection and interpretation: S.F.-J., B.Ö., L.A., E.Å., R.F.-J., and J.A. Sample processing: M.Y., Af.C., Al.C., B.Ö., S.F.-J., F.H., K.L., J.S., L.A., and E.Å. Data generation: M.Y., Af.C., Al.C., W.C., and M.N. Analysis and interpretation of data: M.Y., Af.C., Al.C., A.W., B.Ö., S. F.-J., P.C., G.L., A.L., H.M., R.A.C., S.O., C.S., W.C., J.K., K.L., A.F., and A.S.-S. Preparation of figures: M.Y. and Af.C. Creating schematic illustrations: M.Y. and P.C. Manuscript writing: M.Y. and A.S.-S. Critical revision of the manuscript: all authors.

## Funding

## Competing interests

A.S.-S. is a consultant to Astra-Zeneca on studies not related to the present study. The remaining authors declare no competing interests.

## Additional information

[1]Division of Immunology and Allergy, Department of Medicine Solna, Center for Molecular Medicine, Karolinska Institutet, Karolinska University Hospital, Stockholm, Sweden. [2]Department of Medicine Huddinge, Karolinska Institutet, Stockholm, Sweden. [3]Department of Physiology and Pharmacology, Karolinska Institutet, Stockholm, Sweden. [4]Department of Perioperative Medicine and Intensive Care, Karolinska University Hospital, Stockholm, Sweden. [5]Division of Biostatistics, Institute of Environmental Medicine, Karolinska Institutet, Stockholm, Sweden. [6]The Human Immune Monitoring Center, Institute of Immunity, Transplantation and Infection, Stanford University School of Medicine, Stanford, CA, USA. [7]Division of Infectious Diseases, Department of Medicine Solna, Center for Molecular Medicine, Karolinska Institutet, Stockholm, Sweden. [8]Department of Infectious Diseases, Karolinska University Hospital Solna, Stockholm, Sweden. [9]Department of Microbiology, Tumor and Cell Biology, Karolinska Institutet, Stockholm, Sweden. [10]Clinical Microbiology, Karolinska University Hospital Solna, Stockholm, Sweden. [11]Department of Biochemistry and Biophysics, National Bioinformatics Infrastructure Sweden, Science for Life Laboratory, Stockholm University, Solna, Sweden. [12]Department of Biomedical and Clinical Sciences, Linköping University, Linköping, Sweden. ✉e-mail: anna.smed.sorensen@ki.se

