## [Peer Review File · Nature Communications]

Delayed generation of functional virus-specific circulating T follicular helper cells correlates with severe COVID-19Editorial Note: Figures have been redacted at the request of the authors.

REVIEWER COMMENTS

Reviewer #1 (Remarks to the Author):

In this study Yu et al. have examined circulating T follicular helper (cTfh) cells in patients with COVID-19. Activated cTfh cells were found to correlate with SARS-CoV-2 Spike or RBD activated cTfh cells, the latter being identified by in vitro stimulation analyses. Overall activated cTfh cells correlated with levels of plasmablasts and SARS-CoV-2 antibodies. The development of virus-specific Tfh cells was delayed in severe disease.

Overall, these are interesting studies and are generally well executed. cTfh cells have been defined largely following the published results of Ueno and colleagues. The authors should compare their results closely with the studies reported by Nelson and colleagues (PMID: 35857584) who used peptide-MHC tetramers to identify SARS-CoV-2 specific cTfh cells. They should, in the revised Discussion consider how their results agree with and sometimes seem to disagree with some of the results of that study.

Reviewer #2 (Remarks to the Author):

This is a clearly written manuscript describing a well designed study. The conclusions drawn are reasonable and are supported by the data.

The "weakness" of the manuscript, is that the difficulties of a) only having access to peripheral blood and b) working with limited quantities of patient samples mean that the mechanism underlying the delayed generation of cTfh remains unclear, although some clues to one possible cause are presented. Further, the avalanche of COVID-19 research also means that some of the data presented overlap with already published work.

Nevertheless, it seems to me that this work is a valuable contribution and I would recommend publication.

Reviewer #3 (Remarks to the Author):

In Yu et al. the authors assess the differentiation and functional capacity of circulating Tfh cells in SARS-CoV-2 infected patients. The authors recruited 49 infected patients (and 20 healthy controls) and assessed cTfh cells over time. The authors find that in the acute phase the frequency of total cTfh cells was lower. The frequency of CD38+ICOS+ activated cTfh cells is initially lower (within the first two weeks) but then is higher later on in the acute phase in severe disease patients. This increase correlated with increased serology for Spike and RBD specific antibody. Furthermore, the authors find that the frequency of Spike and RBD specific cTfh cells was increased in moderate, and more so in severe, patients. Lastly, the authors perform functional experiments and show that cTfh cells from severely infected patients support better B cell expansion and antibody production, possibly through increased production of IL-21. Overall this study is presented well and scientific rigor is high. Since alterations in Tfh cells have been implicated in controlling antibody responses to SARS-CoV-2 vaccination and infection, understanding the Tfh response in more detail is of high importance. Although a number of studies have already explored the phenotypes of Tfh cells and their correlation to disease parameters, this study explores the area in more depth. However, a number of substantial issues limit the significance of the study. These include missing controls, validation of assays, and presenting alternative hypotheses.

Major Points

1. Total numbers of cells should be presented in Fig. 1d, 3b and 5b. Since the total frequency of Tfh

cells decreases, and the activated frequency increases, it is difficult to determine whether patients with severe disease had lower, higher or unchanged total numbers of activated Tfh.

2. Fig. 2c should present data for the frequency of each Tfh subset with serological responses, not only the percent activated of the subset.

3. Usage of CD25 as an activation marker is somewhat concerning, as a population of CXCR5+CD25+Foxp3+ Tfr cells may be captured using this. Assessing the frequency of Tfr cells before stim in the samples would be necessary to determine the possible frequency of these cells.

4. The use of full protein for the Tfh stim assays is not optimal since it would require processing and presenting of the antigen for the Tfh AIM assay to work. An alternative explanation for the data is that pbmc from severe patients are able to process and present antigen better. A subset of data should be validated with CoV2 peptide pools which do not require processing and presentation to identify specific cells.

5. In Fig. 3c, does convalescent serological data also correlate with antigen specific Tfh cells?

6. Fig. 3e legend states a line for healthy control responses, but that line is not represented on the plots.

7. Fig. 4 is somewhat overinterpreted because the timing of when the samples were taken is very different, with an average of day 11 for mild and day 34 for severe disease. It is possible that the severe samples have more memory Tfh because of the timing, not because severe disease generates altered Tfh cells. This should be included in the discussion. In addition, it is unclear whether the circulating B cells from severe patients are more primed which can explain the data. A head to head comparison of mild vs. severe Tfh with allogeneic mild B cells would help substantiate authors claims.

8. In Figure 5, data from repeated draws from the same individual should be taken out of the analyses.

RESPONSE TO REVIEWERS

REVIEWER #1:

In this study Yu et al. have examined circulating T follicular helper (cTfh) cells in patients with COVID-19. Activated cTfh cells were found to correlate with SARS-CoV-2 Spike or RBD activated cTfh cells, the latter being identified by in vitro stimulation analyzes. Overall activated cTfh cells correlated with levels of plasmablasts and SARS-CoV-2 antibodies. The development of virus-specific Tfh cells was delayed in severe disease.

Overall, these are interesting studies and are generally well executed. cTfh cells have been defined largely following the published results of Ueno and colleagues. The authors should compare their results closely with the studies reported by Nelson and colleagues (PMID: 35857584) who used peptide-MHC tetramers to identify SARS-CoV-2 specific cTfh cells.

Authors' response: We thank the reviewer for the encouraging comments and appreciate the suggestion to compare our results with the studies by Nelson and colleagues (PMID: 35857584). They utilized an elegant peptide:MHCII tetramer-based strategy to identify SARS-CoV-2 S- and N-specific CD4 T cells in PBMCs in longitudinal samples from convalescent individuals who experienced mild (non-hospitalized) or moderate to severe (hospitalized) COVID-19 in the first wave of the COVID-19 pandemic in Boston, MA, USA. The study focused on HLA-DRB1*07:01 (DR7) positive convalescent subjects and determined the frequency of CD4 T cells recognizing two nonoverlapping S (S166-177 and S310-320) and two nonoverlapping N (N305-316 and N329-340) peptides in the context of DR7. They observed that almost all convalescent individuals had expanded populations of T cells recognizing SARS-CoV-2 epitopes compared to prepandemic controls. There was a trend toward increased frequency of tetramer-positive cells in previously hospitalized patients compared to non-hospitalized patients, however this difference was not statistically significant. Overall the levels of virus-specific CD4 T cells remained stable for up to 10 months in both previously hospitalized and non-hospitalized subjects. Furthermore, when they analyzed the phenotype of the tetramer-positive CD4 T cells, they found that in individuals who had experienced mild COVID-19 virus-specific CD4 T cells displayed higher proportion of cTfh cell phenotype (CXCR5+) compared to hospitalized subjects and that this phenotype was stable over time.

As the reviewer points out there are similarities but also differences between the data published in Nelson *et al.* and the observations presented in our manuscript. These differences may partly relate to the different methods used to identify virus-specific CD4 T cells (tetramers vs. the activation induced marker (AIM) assay). Similar to Nelson *et al.*, we also find stable levels of virus-specific cTfh cells during convalescence using the AIM assay. However, in contrast to what Nelson *et al.* report, we consistently observe that individuals with severe COVID-19 had higher frequencies of virus-specific cTfh cells during convalescence than individuals with mild COVID-19, which is in line with the observation by Juno *et al.* (PMID: 32661393, Extended Data Fig.10 a-b. This paper is cited as reference 26 in the manuscript). This may partly relate to how disease severity was defined. We did not use hospitalization as a means to stratify patients by disease severity, instead we calculated

a respiratory sequential organ failure assessment (SOFA) score providing a higher resolution of disease severity between the mild, moderate and severe patients. When we analyze samples early during acute infection, we find that individuals with mild COVID-19 display higher frequencies of virus-specific cTfh cells compared to patients with severe COVID-19. This is more in line with what Nelson *et al.* find also during convalescence.

The study by Nelson and colleagues centers around the advantage of identifying virus-specific CD4 T cells using tetramers and the phenotype of these cells, including CXCR5 expression associated with cTfh cells. Our study is more focused only on cTfh cells as determined by phenotype and functionality in response to viral proteins. While these approaches should lead to the same conclusion, it may be necessary to use the methods side-by-side on the same samples to correctly compare them. Differences in observations between the studies may also relate to differences in when individuals were sampled; we sampled both during acute, symptomatic infection and longitudinally during convalescence.

We agree with the reviewer that tetramer staining is a powerful method to analyze untouched antigen-specific T cells. However, the AIM assay has the benefit of not being restricted to analyze individuals of a certain HLA type and in response to specific viral peptides. If we had the opportunity to HLA-type our patients, we would have liked to do tetramer stainings where possible, in addition to the AIM assay we used here, to characterize the virus-specific T cell responses to SARS-CoV-2 in even greater detail. In future studies, it would be very interesting and important to compare tetramer and AIM assay data side by side in longitudinal sample sets across disease severity. In the revised version of the manuscript, we have edited the discussion on page 18 to include the following sentences:

“It should be noted that in the current study, virus-specific cTfh cells were identified using an activation induced marker (AIM) assay. Importantly, a study by Nelson and colleagues utilized a peptide:MHCII tetramer-based strategy to identify virus-specific cTfh cells and also observed that SARS-CoV-2-specific cTfh cells persisted several months after recovering from the acute infection⁵⁰, which is in line with the observation in our study. This suggests that there is long term maintenance of SARS-CoV-2-specific cTfh cells after COVID-19. However, Nelson and colleagues observed that a higher proportion of the virus-specific CD4 T cells displayed a higher proportion a cTfh cell phenotype (CXCR5+) in convalescent samples from previously non-hospitalized subjects compared to samples from previously hospitalized subjects. In contrast, we observed that cTfh cells from individuals who recovered from mild COVID-19 displayed a lower proportion of virus-specific cells compared to individuals who recovered from severe COVID-19 during both 3 and 8 months of convalescence. These differences may relate to differences in the methods used, or time of sampling and/or definition of COVID-19 disease severity. Future studies should compare tetramer stainings with the AIM assay side by side in longitudinal sample sets across disease severity to address this.”

REVIEWER #2:

This is a clearly written manuscript describing a well designed study. The conclusions drawn are reasonable and are supported by the data.

The "weakness" of the manuscript, is that the difficulties of a) only having access to peripheral blood and b) working with limited quantities of patient samples mean that the mechanism underlying the delayed generation of cTfh remains unclear, although some clues to one possible cause are presented. Further, the avalanche of COVID-19 research also means that some of the data presented overlap with already published work. Nevertheless, it seems to me that this work is a valuable contribution and I would recommend publication.

Authors' response: We thank the reviewer for the encouraging comments.

REVIEWER #3:

In Yu et al. the authors assess the differentiation and functional capacity of circulating Tfh cells in SARS-CoV-2 infected patients. The authors recruited 49 infected patients (and 20 healthy controls) and assessed cTfh cells over time. The authors find that in the acute phase the frequency of total cTfh cells was lower. The frequency of CD38+ICOS+ activated cTfh cells is initially lower (within the first two weeks) but then is higher later on in the acute phase in severe disease patients. This increase correlated with increased serology for Spike and RBD specific antibody. Furthermore, the authors find that the frequency of Spike and RBD specific cTfh cells was increased in moderate, and more so in severe, patients. Lastly, the authors perform functional experiments and show that cTfh cells from severely infected patients support better B cell expansion and antibody production, possibly through increased production of IL-21. Overall this study is presented well and scientific rigor is high. Since alterations in Tfh cells have been implicated in controlling antibody responses to SARS-CoV-2 vaccination and infection, understanding the Tfh response in more detail is of high importance. Although a number of studies have already explored the phenotypes of Tfh cells and their correlation to disease parameters, this study explores the area in more depth. However, a number of substantial issues limit the significance of the study. These include missing controls, validation of assays, and presenting alternative hypotheses.

Major Points

Point 1. Total numbers of cells should be presented in Fig. 1d, 3b and 5b. Since the total frequency of Tfh cells decreases, and the activated frequency increases, it is difficult to determine whether patients with severe disease had lower, higher or unchanged total numbers of activated Tfh.

Authors' response: We agree with the reviewer that displaying cell frequencies is a limitation of the study and makes it difficult to determine whether the absolute/total number of cTfh cells are altered across disease severity and time. Unfortunately, as mentioned in the discussion, in the majority of our severe and moderate patients, differential cell counts to determine absolute lymphocyte counts were not done on the same day as the samples were collected for this study, thus we could not accurately calculate the absolute number of cTfh cells in the blood at every sampling point.

However, in a limited number of patient samples (n=5 mild, n=8 moderate and n=7 severe), we do have differential cell counts available from the same sampling time as we performed flow analysis to measure cTfh cells in the blood. Using the differential cell count data, we calculated the absolute numbers of total cTfh cells, activated and virus-specific cTfh cells, displayed in **Reviewer figure 1**. In line with what we observed for frequencies of total cTfh cells in **Figure 1d**, the absolute numbers of total cTfh cells in patients with severe disease were significantly lower than patients with mild disease during acute SARS-CoV-2 infection (**Reviewer figure 1a**). Similar as with activated cTfh cell frequencies, we also observed higher absolute numbers of activated cTfh cells in patients with severe disease compared to patients with mild disease (**Reviewer figure 1b**). However, these differences were not statistically significant, likely due to the limited sample number in each group. **Reviewer**

figure 1c-d display the absolute numbers of Spike and RBD-specific cTfh cells, respectively across disease severity in acute disease. Again there are no statistically significant differences between groups with different disease severity, likely due to limited sample size. However, the tendency is similar to what we observed in main **Figure 3b** with frequencies of virus-specific cTfh cells, suggesting that patients with severe disease display higher absolute numbers of virus-specific cTfh cells compared to mild patients. In addition, we also observed that the absolute numbers of cTfh cells correlated well with the frequencies of cTfh cells, for total (bulk) cTfh cells (**Reviewer figure 1e**), activated cTfh cells (**Reviewer figure 1f**) and Spike-specific cTfh cells (**Reviewer figure 1g**) and RBD-specific cTfh cells (**Reviewer figure 1h**). Unfortunately, we lack the data to display longitudinal changes in absolute numbers of cTfh cells across disease severity as displayed in **Figure 5b-c**.

Still, as shown in **Reviewer figure 1**, the data we do have available on absolute cTfh cell counts display a similar pattern as shown when frequencies of cTfh cells are plotted. We could add **Reviewer figure 1** as a supplementary figure to the revised manuscript, if the editor and reviewer(s) determine that it strengthens the study.

Editorial note: figure redacted

Point 2. Fig. 2c should present data for the frequency of each Tfh subset with serological responses, not only the percent activated of the subset.

Authors' response: We have plotted the frequencies of each cTfh subset with serological IgA and IgG responses against SARS-CoV-2 Spike and RBD, respectively in **Reviewer figure 2a and b**. We have added this figure as a new **Supplementary figure 2** in the revised version of the manuscript.

In contrast to what we observed in **Figure 2c** with activated cTfh cells, we did not find any correlations between total frequencies of each cTfh cell subset and antibody responses. However, we do not find this too surprising as the virus-specific antibody responses likely primarily associate with the activated or virus-specific cTfh cell responses, and not necessarily the frequencies of total cTfh cells (or subsets). As mentioned in the discussion, frequencies of activated and virus-specific cTfh cells correlate (**Reviewer figure 2 / Supplementary figure 2**), suggesting that activated cTfh cells may reflect recent or ongoing virus encounter and emigration of cTfh cells from the GCs. In the revised version of the manuscript, we have included the new **Supplementary figure 2** and also commented on this figure on page 8 in the results section.

Editorial note: figure redacted

Point 3. Usage of CD25 as an activation marker is somewhat concerning, as a population of CXCR5+CD25+Foxp3+ Tfr cells may be captured using this. Assessing the frequency of Tfr cells before stim in the samples would be necessary to determine the possible frequency of these cells.

Authors' response: We agree with the reviewer that using CD25 as an activation marker may risk including CXCR5+CD25+Foxp3+ Tfr cells in the analysis. To test this, we analyzed the frequencies of CD25+ Foxp3+ expressing Treg cells in the CXCR5+ memory CD4 T cell gate in biobanked samples collected from COVID-19 patients during acute infection across disease severity before stimulation. We found the frequencies of CD25+ Foxp3+ CXCR5+ cTfr cells to be very low or undetectable in the majority of samples tested (**Reviewer figure 3a-b**). These data suggest that CD25+ Foxp3+ CXCR5+ cTfr cells are so low in frequency in the CXCR5 gate that they would not contribute significantly to frequencies of CD25 expressing cTfh cells after stimulation with viral antigen.

Editorial note: figure redacted

Point 4. The use of full protein for the Tfh stim assays is not optimal since it would require processing and presenting of the antigen for the Tfh AIM assay to work. An alternative explanation for the data is that pbmc from severe patients are able to process and present antigen better. A subset of data should be validated with CoV2 peptide pools which do not require processing and presentation to identify specific cells.

Authors' response: The reviewer correctly points out that full protein requires processing into peptides and subsequent presentation of peptide antigen, which may be impacted by alterations in the antigen-presenting cells across the patient samples. As suggested, we have validated our cTfh cell stimulation data using full-length protein by comparing stimulation side-by-side with SARS-CoV-2 Spike peptide in a subset of patient samples from mild, moderate and severe COVID-19 patients (**Reviewer figure 4a**). PBMCs from COVID-19 patients were cultured and stimulated with 5µg/mL SARS-CoV-2 spike peptide pool (15mers overlapping by 11 spanning the full-length Spike protein) or full-length Spike protein for 20 hours. PBMCs were also stimulated with 5µg/mL myelin oligodendrocyte glycoprotein (MOG) peptide pool as negative control, and 0.1 µg/mL SEB as positive control. Virus-specific cTfh cells were assessed by determining the frequencies of CD25+CD134+ cells. We found similar frequencies of spike-specific cTfh cells in response to spike peptide and spike protein stimulation (**Reviewer figure 4b**). Importantly, the pattern of higher frequencies of spike-specific cTfh cells in severe patients compared to moderate compared to mild patients was similar irrespective of whether we used peptide or protein as the source of antigen. The differences across disease severity were not statistically significant likely due to the limited number of biobanked samples available for this validation experiment. Furthermore, we found that frequencies of spike-specific cTfh cells assessed using peptide correlated well with the frequencies of spike-specific cTfh cells assessed by spike protein (**Reviewer figure 4c**). Together, these data further suggest the increase in frequencies of virus-specific cTfh cells in more severe COVID-19 patients during acute infection is not a function of altered processing and presentation capacity.

Editorial note: figure redacted

Point 5. In Fig. 3c, does convalescent serological data also correlate with antigen specific Tfh cells?

Authors' response: Indeed, the frequencies of antigen-specific cTfh cells during acute SARS-CoV-2 infection displayed in **Figure 3c** also correlated with serological data at 3 and 8 months convalescence, as shown below in **Reviewer figure 5**. This data could be included in the revised manuscript as an additional supplementary figure, if the reviewer and editor think it merits the manuscript.

Editorial note: figure redacted

Point 6. Fig. 3e legend states a line for healthy control responses, but that line is not represented on the plots.

Authors' response: **Figure 3e** actually does have a line depicting the cytokine levels from supernatants of healthy control PBMCs stimulated with SARS-CoV-2 spike and RBD protein. However, the cytokine levels in healthy control PBMCs are so low or even undetectable that they are difficult to display properly. Below we have inserted a modified version of the same data as **Reviewer figure 6**, which is a magnification of the graphs shown in **Figure 3e** of the manuscript. We hope that this more clearly shows the dotted line referred to in the figure legend.

Editorial note: figure redacted

Point 7. Fig. 4 is somewhat overinterpreted because the timing of when the samples were taken is very different, with an average of day 11 for mild and day 34 for severe disease. It is possible that the severe samples have more memory Tfh because of the timing, not because severe disease generates altered Tfh cells. This should be included in the discussion. In addition, it is unclear whether the circulating B cells from severe patients are more primed which can explain the data. A head to head comparison of mild vs. severe Tfh with allogeneic mild B cells would help substantiate authors claims.

Authors' response: We agree with the reviewer that it is unfortunate that the time from symptom onset of sampling the mild and severe COVID-19 patients for cell subset sorting and co-culture experiments shown in **Figure 4** and **Supplementary table 5** were different. Indeed, in the samples used for the experiments in Figure 4, we did observe higher frequencies of activated cTfh cells in the samples from COVID-19 patients with severe disease compared to those from patients with mild disease (**Supplementary figure 8**). However, we argue that the data subsequently shown in **Figure 5** would support that the differences in frequencies of activated or virus-specific memory cTfh cells in patients with mild vs. severe disease, mainly stem from disease severity rather than timing of sampling. In **Figure 5**, we see differences in frequencies of activated or virus-specific memory cTfh cells across disease severity when comparing the same time after symptom onset. However, since we did not perform functional assessment on all the samples displayed in **Figure 5**, we agree with the reviewer that we have to be cautious not to overinterpret the data presented in **Figure 4**. In the revised version of the manuscript, we have further emphasized the impact that the time of sampling may have on the results by adding the following sentence to the discussion on page 20:

“The difference in sampling time after symptom onset of mild and severe COVID-19 patients for samples used for functional assays (Supplementary Table 5) likely contributed to a difference in frequencies of activated cTfh cells”.

We agree with the reviewer that if the circulating B cells from patients with severe disease are more primed, this could explain the data as well. However, data from a head-to-head comparison of Tfh cells isolated from patients with mild vs. severe disease co-cultured with allogeneic B cells from a mild COVID-19 patient may not be so easy to interpret, since the mismatched MHC (allo-) response likely would be both different between the patients. In **Figure 4**, we coculture cTfh cell with autologous naïve B cells isolated from COVID-19 patients, which should not be primed in either severe or mild COVID-19 patients. The higher generation of plasmablast and antibodies were still observed in samples from severe patients compared to mild. These data highlight the functionality of cTfh cells instead of primed B cell contributions to the observation. Unfortunately, all patient samples have been used up to do the *in vitro* coculture experiments in **Figure 4** so it is not feasible to test the isolated memory B cell priming in these samples. Currently it is not possible to recruit new COVID-19 patients with the same SARS-CoV-2 strain infection without vaccination as the cohort in this study, to further test this. We agree that the lack of B cell priming data in **Figure 4** is a limitation of our study and we appreciate reviewer pointing this out. We have further emphasized the potential impact that B cell priming may have on the results by adding the following sentence to the discussion on page 20:

“Unfortunately, we did not compare the priming of naïve and memory B cells isolated from COVID-19 patients, which likely contributed to a difference in frequencies of differentiated plasmablast, which is a limitation of this study.

*Due to the limited blood volume we were able to obtain from each COVID-19 patient, it was not feasible to isolate sufficient numbers of activated vs. non-activated cTfh cells, or SARS-CoV-2-specific vs. non-specific cTfh cells for subsequent co-culture with **autologous B cells primed the same way**, which is also a limitation of this study.”*

Point 8. In Figure 5, data from repeated draws from the same individual should be taken out of the analyzes.

Authors’ response: Study statistician and co-author, Anna Warnqvist, has helped word this response. We politely disagree with the reviewer that we should remove data points from the analysis shown in **Figure 5**. Even though most standard statistical methods assume the independence of observations, there are currently several established and reliable statistical methods available to handle correlated samples (Generalized estimation equations, multilevel mixed models, the fixed effects model and different variants of clustered robust standard error estimators, for example). We would argue for the greater need to implement such methods. Reducing the sample artificially by choosing the samples analyzed will always reduce power and at worst can induce bias. As presented in the Methods Statistical Analysis session, for our analysis we chose to use Generalized estimating equations (GEE) to account for the intra-person correlations inherent to repeated measures. We take this opportunity to highlight that in **Figure 5**, samples from the same individual at different time points during acute SARS-CoV-2 infection were deliberately linked with line to clarify this inherent data interrelatedness.

REVIEWERS' COMMENTS

Reviewer #1 (Remarks to the Author):

No new comments

Reviewer #3 (Remarks to the Author):

The authors have addressed most of my concerns.

For point 3, it is unclear why the Tfr gate was drawn where it is when there is a clear CD25+FoxP3+ population not gated. Do CXCR5- Treg fall in the drawn CD25+FoxP3+ gate? This data should also be added to supplement.

RESPONSE TO REVIEWERS

REVIEWER #1:

No new comments

REVIEWER #3:

The authors have addressed most of my concerns.

For point 3, it is unclear why the Tfr gate was drawn where it is when there is a clear CD25+FoxP3+ population not gated. Do CXCR5- Treg fall in the drawn CD25+FoxP3+ gate? This data should also be added to supplement.

Authors' response: Based on the suggestion of the reviewer, we have revised the gating strategy to ensure that we include all CD25+FoxP3+ cells (**Reviewer figure 1a**). The more generous gate does not alter the conclusion of the analysis. We still found the frequencies of CD25+ Foxp3+ CXCR5+ cTfr cells to be very low or undetectable in the majority of samples (**Reviewer figure 1a-b**). These data suggest that CD25+ Foxp3+ CXCR5+ cTfr cells are so low in frequency in the CXCR5 gate that they would not contribute significantly to frequencies of CD25 expressing cTfh cells after stimulation with viral antigen.

As prompted by the reviewer, we now also analyzed whether CXCR5- Treg cells fall into the CD25+FoxP3+ gate (**Reviewer figure 1a**). We observed similar distribution of Treg cells in the CD25+FoxP3+ gate as Tfr (**Reviewer figure 1a**). In addition, we found overall, both CD25+ Foxp3+ CXCR5+ cTfr cells and CD25+ Foxp3+ CXCR5- Treg cells are relatively low in frequency in majority of the patient samples (**Reviewer figure 1c**).

We agree with the reviewer that these data should be added to supplement. We have added a new supplementary figure 3 in the revised manuscript, to show the CXCR5+CD25+FoxP3+ population in PBMCs of COVID-19 patients before viral antigen stimulation. We also state in the results that CD25+ Foxp3+ CXCR5+ cTfr cells are so low in frequency in the CXCR5 gate that they would not contribute significantly to frequencies of CD25 expressing cTfh cells after stimulation with viral antigen. As this manuscript is focused on cTfh cells, we did not include the Treg data in revised supplementary figure 3.

Editorial note: figure redacted